# Negative regulation of G2-M by ATR (mei-41)/Chk1(Grapes) facilitates tracheoblast growth and tracheal hypertrophy in Drosophila

Amrutha Kizhedathu[1,2†], Archit V Bagul[1†], Arjun Guha[1]*

[1]Institute for Stem Cell Biology and Regenerative Medicine, Bangalore, India; [2]SASTRA University, Thanjavur, India

**Abstract** Imaginal progenitors in Drosophila are known to arrest in G2 during larval stages and proliferate thereafter. Here we investigate the mechanism and implications of G2 arrest in progenitors of the adult thoracic tracheal epithelium (tracheoblasts). We report that tracheoblasts pause in G2 for ~48–56 h and grow in size over this period. Surprisingly, tracheoblasts arrested in G2 express drivers of G2-M like Cdc25/String (Stg). We find that mechanisms that prevent G2-M are also in place in this interval. Tracheoblasts activate Checkpoint Kinase 1/Grapes (Chk1/Grp) in an ATR/mei-41-dependent manner. Loss of ATR/Chk1 led to precocious mitotic entry ~24–32 h earlier. These divisions were apparently normal as there was no evidence of increased DNA damage or cell death. However, induction of precocious mitoses impaired growth of tracheoblasts and the tracheae they comprise. We propose that ATR/Chk1 negatively regulate G2-M in developing tracheoblasts and that G2 arrest facilitates cellular and hypertrophic organ growth.
DOI: https://doi.org/10.7554/eLife.29988.001

*For correspondence:
arjung@instem.res.in

†These authors contributed equally to this work

Competing interests: The authors declare that no competing interests exist.

## Introduction

The precise regulation of cell division in space and time is essential for normal development. This is achieved by the integration of developmental signals and the machinery that drives cell cycle progression. Non-cycling cells are known to pause in G1 (or G0) (*Cheung and Rando, 2013*) or, less commonly, in the G2 phase (*Bouldin and Kimelman, 2014*) of the cell cycle and rekindle a mitotic program at the appropriate time and location. G2-arrested cells have been observed in developing Drosophila (*Johnston and Edgar, 1998*; *Ayeni et al., 2016*), Ciona (*Ogura et al., 2011*), zebrafish (*Nguyen et al., 2017*), chick (*Boije et al., 2009*; *Stone et al., 1999*) and mice (*Seki et al., 2007*). In this study we investigate the cell-intrinsic mechanisms underlying developmental G2 arrest and the significance of this mode of arrest in the context of the Drosophila tracheal (respiratory) system.

Fruit flies undergo a complete metamorphosis during their life cycle and have distinct larval and adult body forms. Progenitors of the adult animal (hereafter referred to as imaginal progenitors) are set aside during embryonic development, nourished in the larva, and activated during the larval-adult transition (*Cohen et al., 1993*). These progenitors generate adult organs de novo (sensory organs, wings, legs) or by remodeling larval organs (gut, abdominal epidermis, tracheal system). Imaginal progenitors of different tissues may either proliferate throughout larval life or in a punctuated manner. The progenitors of the abdominal epidermis (histoblasts)(*Ninov et al., 2009*), and the thoracic tracheal (respiratory) system (*Djabrayan et al., 2014*; *Guha et al., 2008*; *Sato et al., 2008*; *Weaver and Krasnow, 2008*; *Pitsouli and Perrimon, 2010*) exhibit the latter pattern. These cells remain mitotically arrested in G2 through most of larval life and initiate a program of rapid cell division and morphogenesis thereafter.

**eLife digest** Every organism begins as a single cell. That cell, and all the other cells it generates over time, need to divide at the right time and in the right place to develop into an adult. As they do so, they pass through the stages of the cell cycle. As cells prepare to divide they enter into the first growth phase, G1, ramping up their metabolic activity. They then enter S phase, duplicate their DNA, and subsequently a second growth phase G2. Finally, during the mitotic phase, the chromosome separate and cells undergo cytokinesis to form new cells.

Dividing cells can pause at certain stages of the cell cycle to assess whether the conditions are suitable to proceed. The length of the pause depends on the stage of development and the cell type. Signals around the cell provide the cues that it needs to make the decision.

The fruit fly *Drosophila melanogaster*, for example, undergoes metamorphosis during development, meaning it transforms from a larva into an adult. The larva contains small patches of 'progenitor' cells that form the adult tissue. These remain paused for various intervals during larval life and restart their cell cycle as the animal develops. A key challenge in biology is to understand how these progenitors pause and what makes them start dividing again. Here, Kizhedathu, Bagul and Guha uncover a new mechanism that pauses the cell cycle in developing animal cells.

Progenitors of the respiratory system in the adult fruit fly pause at the G2 stage of the cell cycle during larval life. Some of these progenitors, from a part of the larva called the dorsal trunk, go on to form the structures of the adult respiratory system. By counting the cells and tracking their dynamics with fluorescent labels, Kizhedathu et al. revealed that the progenitor cells pause for between 48 and to 56 hours. Previous research suggested that this pause happens because the cells lack a protein essential for mitosis called Cdc25/String. However, these progenitors were producing Cdc25/String. They stopped dividing because they also made another protein, known as Checkpoint Kinase 1/Grapes (Chk1/Grp).

Chk1 is known to add a chemical modification to Cdc25, which dampens its activity and stops the cell cycle from progressing. This is likely what allow the flies to co-ordinate their development and give the cells more time to grow. When Chk1 was experimentally removed, it reactivated the paused cells sooner, resulting in smaller cells and a smaller respiratory organ.

This work extends our understanding of stem cell dynamics and growth during development. Previous work has shown that cells that give rise to muscles and the neural tube (the precursor of the central nervous system) also pause their cell cycle in G2. Understanding more about how this happens could open new avenues for research into developmental disease.

DOI: https://doi.org/10.7554/eLife.29988.002

The mechanism for G2-M is highly conserved among eukaryotic cells (*Bouldin and Kimelman, 2014*). The G2-M transition is triggered by the dephosphorylation of the Cyclin-dependent Kinase Cdc2/Cdk1 by the phosphatase Cdc25. Dephosphorylation of Cdc2/Cdk1 by Cdc25 results in the activation of the Cdk-Cyclin B complex and in turn to the phosphorylation of substrates in the cytoplasm and nucleus and to mitotic entry. G2-arrested progenitors of sensory organs (SOPs) in Drosophila have been shown to repress Cdc25/String (Stg) expression (*Johnston and Edgar, 1998*). A recent study showed that SOPs forced to divide precociously undergo normal neuronal differentiation but generate supernumerary non-neuronal sensory organ support cells. Taken together, studies on SOPs show that G2 arrest is mediated by the transcriptional repression of Stg and that the arrest ensures that a proper balance of cell types within each sensory organ is achieved (*Ayeni et al., 2016*).

Whether other G2-arrested progenitors in Drosophila and in other organisms are regulated like SOPs is an open question. Histoblasts appear to be regulated in a similar manner. Non-cycling histoblasts lack Stg expression and upregulate Stg in an ecdysteroid signaling-dependent manner during G2-M(*Ninov et al., 2009*). In the sea squirt Ciona, progenitors undergoing neurulation are known to arrest in G2 (*Ogura et al., 2011*). The overexpression of Stg in these cells triggers precocious G2 exit and perturbs the formation of the neural tube (*Ogura et al., 2011*). This suggests that G2-arrested neural progenitors in Ciona may be regulated in a manner akin to SOPs. A recent study on the muscle stem cells that contribute to the growth of the zebrafish myotome has shown that these

cells arrest in G2 but utilize a different mechanism for arrest. G2-arrest in this context is mediated by Meox1-dependent repression of Cyclin B expression (*Nguyen et al., 2017*).

Proliferating cells in the G2 phase that are subjected to DNA damaging agents stall cell cycle progression, initiate DNA repair, and rekindle the cell cycle after repair is completed (*Branzei and Foiani, 2008*). Cells with DNA damage utilize other mechanisms for G2 arrest. Genotoxic stress leads to the activation of the phosphoinositide-3-kinase-related kinases ATR and ATM that phosphorylate and activate Checkpoint Kinases 1 and 2 (Chk1,2) respectively(*Kumagai et al., 2004*; *Chaturvedi et al., 1999*). Chk1 and Chk2, in turn, inhibit Cdc25 and arrest cell cycle progression (*Xiao et al., 2003*; *Chaturvedi et al., 1999*). In addition, Chk1 can also stabilize the Wee/Myt kinases that phosphorylate and inhibit Cdc2/Cdk1(*O'Connell et al., 1997*). While there is evidence that ATR/Chk1 can regulate cell cycle progression in cultured cells in the absence of induced DNA damage (*Sørensen et al., 2004*; *Tang et al., 2006*) and during early embryonic development (*Sibon et al., 1999*; *Sibon et al., 1997*; *Su et al., 1999*; *Liu et al., 2000*) their roles in developmental G2 arrest have not been fully explored.

In this study we focus on the progenitors of the adult thoracic tracheal system (tracheoblasts). It has been reported that tracheoblasts arrest in G2 during early larval life and rekindle a mitotic program at the onset of the pupal period. Our efforts to determine precisely when tracheoblasts enter and exit G2 showed that the cells enter larval life in G1, transition from G1 to G2 in the first larval instar (L1) and remain in G2 till the mid third larval instar (L3), a period of ~48–56 h, whereupon they enter mitosis. We investigated the status of Stg expression in G2-arrested tracheoblasts to find that these cells express Stg throughout. Moreover, we observed that paused tracheoblasts also expressed Cdc2/Cdk1 and Cyclin B. These findings alerted us to the possibility that G2 arrest is not mediated via transcriptional repression of Stg, or of the essential drivers for G2-M, and led us to investigate alternate mechanisms. The findings presented here show that the G2-M transition in tracheoblasts is negatively regulated by ATR (*mei-41*, hereafter ATR)/Chk1 (*Grapes* (*Grp*), hereafter Chk1), that the transition involves the coordination of several processes including the downregulation of Chk1, and that arrest in G2 facilitates cellular and organ growth.

## Results

### Thoracic tracheoblasts arrested in G2 express drivers of G2-M

The tracheal system of the Drosophila larva originates in the embryo from a pair of placodes in each segment. Each placode undergoes branching morphogenesis to generate tracheal tubes that anastomose at stereotyped locations and generate a connected network. The larval tracheal system is largely comprised of cells that undergo repeated cycles of endoreplication and are post-mitotic. In addition to serving the oxygen demands of the larva, the larval tracheal network also serves as a scaffold for the development of the tracheal system of the adult fruit fly. Embedded in larval tracheae, at stereotyped locations, are imaginal tracheal progenitors. These cells remain mitotically quiescent through larval life and proliferate and replace larval cells during metamorphosis (*Pitsouli and Perrimon, 2010*; *Weaver and Krasnow, 2008*). The progenitors that contribute to the development of the thoracic tracheal system of the adult animal, specifically progenitors of the second thoracic metamere (Tr2), are unusual. To the best of our knowledge, majority of the cells that populate Tr2 tracheae during larval stages are also imaginal progenitors (*Figure 1A*, shown in green)(*Guha et al., 2008*; *Sato et al., 2008*). Unlike post-mitotic larval tracheal cells, the cells that comprise Tr2 tracheae are mitotically competent, remain paused in G2 during larval life, and rekindle mitoses in the third larval instar (L3)(*Guha et al., 2008*; *Sato et al., 2008*; *Djabrayan et al., 2014*).

For the analysis of the mechanism for G2 arrest in Tr2 we focused our attention on the cells that comprise the Dorsal Trunk (DT) in this segment (*Figure 1A*, demarcated with dashed lines). Tr2 DT constitutes a developmental compartment consisting of 16–18 cells that become mitotically active mid-L3 (*Rao et al., 2015*). To characterize when precisely cells in Tr2 DT enter and exit G2, we counted the numbers of cells at different larval stages (first instar (L1), second instar (L2), third instar (L3) (early L3, 32–40 h L3, wandering L3 (WL3)) and analyzed the timecourse of BrdU incorporation (S phase) and phospho-Histone H3 (pH3, M phase) labeling at these respective stages (*Figure 1I*, *Figure 1—figure supplement 1*). Counts of cell numbers in Tr2 DT showed that there are 16–18 cells in L1, that this number remains unchanged through 16–24 h L3, that there are 25–35 cells by

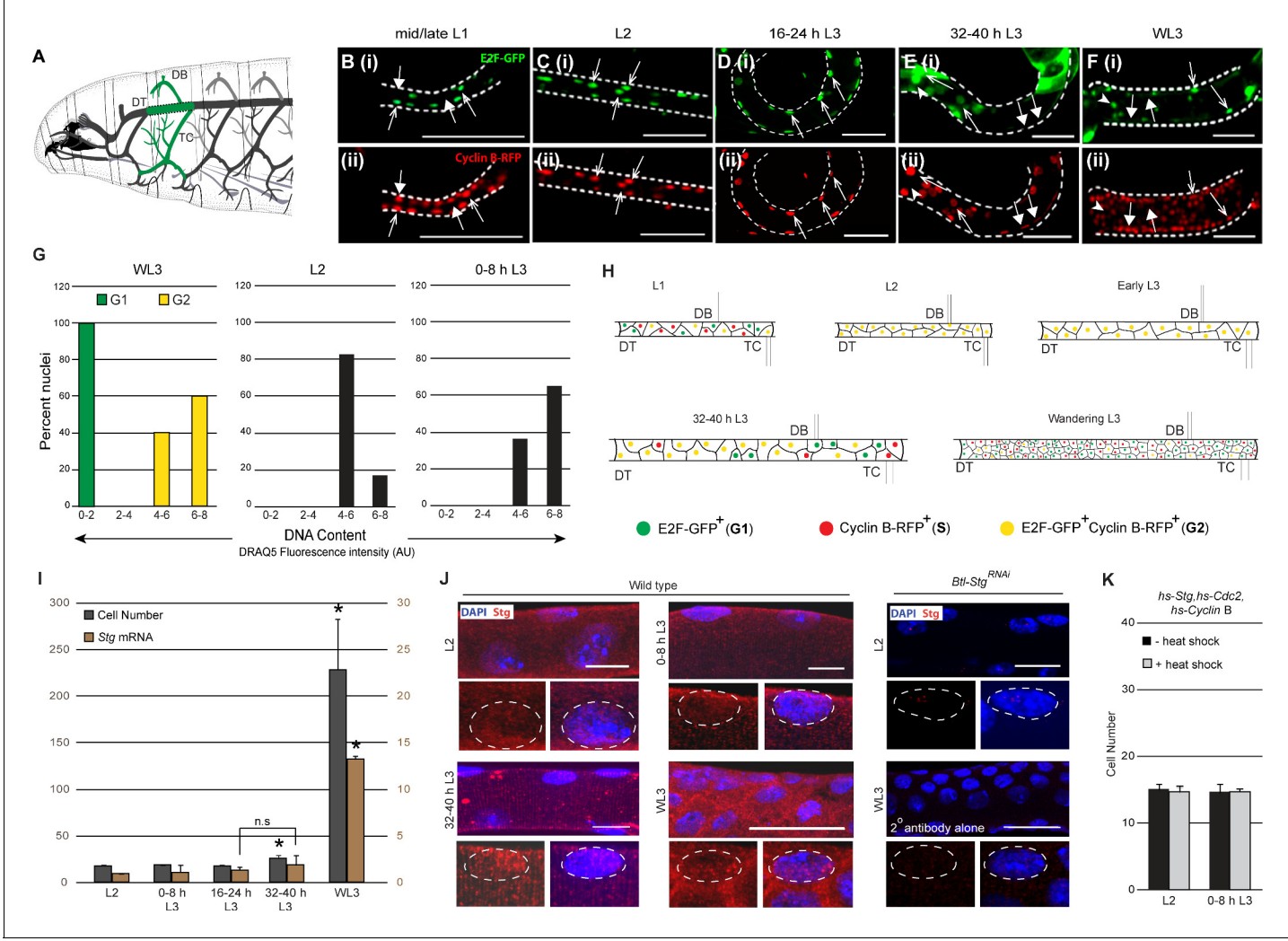

**Figure 1.** Tracheal progenitors arrested in G2 express Cdc25/String. (**A**) Diagram of a third instar larva (L3) showing the tracheal branches in the second thoracic metamere, Tr2 (green). Dashed lines indicate the Tr2 Dorsal Trunk (DT) segment. (**B–F**) Expression of FUCCI reporters E2F1-GFP (green, arrowheads B(i)-F(i)) and CyclinB-RFP (red, filled arrows (B(ii)-F(ii)) at different larval stages. Double-positive cells are indicated with open arrows. Scale bar = 50 µm. (**G**) Comparison of the DNA content of mitotically active cells in G1 and G2 (left panel, cells from Tr2 DT at the wandering L3 stage staged using FUCCI) and G2-paused cells in Tr2 DT at L2 (middle panel) and 0–8 h L3 (right panel). DNA content was estimated using DRAQ5 fluorescence intensity (see methods). Plotted in the histogram are DNA content measurements from WL3 G1 (n = 13 nuclei) WL3 G2 (yellow, n = 15 nuclei), L2 (n = 23 nuclei) and 0–8 h L3 (n = 14 nuclei). (**H**) Cartoon showing the cell cycle program of cells in Tr2 DT at different larval stages. (**I**) Numbers of cells (grey bars) and *Stg* mRNA levels (quantitative PCR (brown bars)) in Tr2 DT at different larval stages. Graph shows cell numbers (mean ± standard deviation, n ≥5 tracheae per timepoint, grey axes) and fold change in mRNA levels with respect to L2 (mean ± standard deviation, brown axes). (**J**) Stg immunostaining (red) in Tr2 DT in wild type at different stages. Also shown are Tr2 DT from *Btl-Stg^RNAi* and from wild type larvae stained with the secondary antibody alone (far right panels). The distribution of Stg in the nucleus and cytoplasm can be seen in the higher magnification views of single nuclei below each panel. (**K**) Effect of heat shock-dependent co-expression of *Stg, Cdc2* and *CyclinB* on cell number in Tr2 DT (mean ± standard deviation, n ≥ 5 tracheae per timepoint). DT = Dorsal Trunk,DB = Dorsal Branch, TC = Transverse Connective.Scale bar = 10 µm. Student's paired t-test: *p<0.05. n.s = not significant.

DOI: https://doi.org/10.7554/eLife.29988.003

The following source data and figure supplements are available for figure 1:

**Source data 1.** *Figure 1G*: Background intensity corrected values for DRAQ5 intensities at L2, 0–8 h L3 and WL3 *Figure 1I*: Numerical data for number of cells in Tr2 DT of wild type larvae at different stages.
DOI: https://doi.org/10.7554/eLife.29988.006

**Figure supplement 1.** Tr2 tracheoblasts enter S phase in L1 and enter M phase mid L3, after a period of ~48–56 h.
DOI: https://doi.org/10.7554/eLife.29988.004

**Figure supplement 2.** Tr2 tracheoblasts arrested in G2 express Cdc2/Cdk1 and Cyclin B.

*Figure 1 continued on next page*

Figure 1 continued

DOI: https://doi.org/10.7554/eLife.29988.005

32–40 h L3 and ~250 cells at wandering L3 (WL3) (*Figure 1I*, n ≥ 5 tracheae per timepoint here and in all subsequent figures showing cell frequencies). BrdU incorporation in Tr2 was observed in the first larval instar (L1) and again at 32–40 h L3 but not in the interim period. Phospho-histone H3 labeling was detected from 32- 40 h L3 onwards (representative images from different stages and a quantitation of pH3$^+$ cells at these stages is shown in *Figure 1—figure supplement 1*). Based on these analyses we surmised that cells in Tr2 DT complete S phase in L1 and remain in G2 from late L1/L2 till 24–32 h L3.

The FUCCI (Fluorescence Ubiquitination-based Cell Cycle Indicator) system facilitates precise cell cycle staging based on the levels of expression of fluorescent reporters (*Zielke et al., 2014*).To characterize cell cycle phasing of cells in Tr2 DT, we expressed fluorescently tagged degrons from E2F1 (E2F1-GFP, degraded at the onset of S phase) and Cyclin B (Cyclin B-RFP, degraded in mitosis) in the tracheal system and analyzed expression of these reporters in Tr2 DT at different larval stages (*Figure 1B–F*, n ≥ 6 tracheae per timepoint analyzed). In this system, cells in G1 are GFP$^+$, cells in S RFP$^+$ and cells in G2 GFP$^+$RFP$^+$. We found that cells in Tr2 DT are heterogeneous with respect to reporter expression in L1 (GFP$^+$,RFP$^+$ and GFP$^+$RFP$^+$) (*Figure 1B*), homogeneous from L2 till early L3 (GFP$^+$RFP$^+$) (L2 shown *Figures 1C*, 0-8 h L3 not shown, 16–24 h L3 shown in *Figure 1D*) and heterogeneous from 32-40 hL3 onwards (GFP$^+$, RFP$^+$ and GFP$^+$RFP$^+$) (32–40 h L3 shown in *Figure 1E*, WL3 shown in *Figure 1F*). Analysis of freshly hatched L1s showed that the cells in Tr2 DT are all GFP$^+$ (*Figure 1—figure supplement 1*). Taken together, FUCCI analysis show that cells in Tr2 enter larval life in G1, transition from G1 to G2 in the first larval instar (L1) and remain paused in G2 from the second larval instar (L2) till 24–32 h L3 (~48–56 h). In addition to these analyses, we also compared the DNA content of paused cells in L2 and early L3 with the DNA content of mitotically active cells in the G1 or G2 phase (G1/G2 identified by FUCCI at WL3) (*Figure 1G*). We noted that the DNA content of cells in L2 and early L3 was comparable to the DNA content of cells in G2. Thus, FUCCI and DNA content analyses corroborated our initial assessment of cell cycle phasing of cells in Tr2 DT during larval life (summarized in the diagram in *Figure 1H*).

Studies in SOPs and other imaginal progenitors have shown that G2 arrest is due to the absence of Stg expression (*Johnston and Edgar, 1998*). To test whether this model is applicable to Tr2 tracheoblasts, we compared levels of *Stg* mRNA in Tr2 DT prior to and post mitotic entry using realtime PCR (qPCR) and immunohistochemistry. For qPCR analysis we micro-dissected Tr2 DT fragments from different stages and isolated RNA from these fragments (≥15 tracheal fragments per timepoint per experiment, n = 3 experiments). Levels of *Stg* mRNA at L2, 0–8 h L3, 16–24 h L3 and 32–40 h L3 were comparable (*Figure 1I*) and the level at WL3 was significantly higher than earlier stages (*Figure 1I*). Stg immunostaining revealed low levels of Stg, in both nucleus and cytosol, at L2, 0–8 h L3 and 32–40 h L3 (*Figure 1J*, n = 9 tracheae per condition per experiment, n = 3 experiments). We detected higher levels, with the same spatial distribution as earlier stages, at WL3 (*Figure 1J*). To confirm that the anti-Stg antibody was specific, we stained trachea from animals expressing Stg$^{RNAi}$ in the tracheal system under the control of *Btl*-Gal4 (*Btl*-Stg$^{RNAi}$). Staining in these animals was comparable to the staining in specimens incubated with the secondary antibody alone indicating that the Stg staining is specific (*Figure 1J*). These experiments suggest that unlike SOPs, tracheoblasts paused in G2 express both *Stg* mRNA and protein. Next we investigated whether Cdc2/Cdk1 and Cyclin B, the other drivers of G2-M, are also expressed in tracheoblasts paused in G2. Both mRNA and protein were detected in paused cells and exhibited a timecourse of expression like *Stg*/Stg (qPCR data and immunostaining shown in *Figure 1—figure supplement 2*, qPCR as above n = 3, immunohistochemistry n = 6 tracheae per condition, n = 2 experiments). Together, the findings suggest that G2-arrested cells in Tr2 DT express all the cell cycle regulators necessary for G2-M.

## ATR/Chk1contribute to G2-arrest in tracheoblasts

The expression of drivers of G2-M in paused cells was unexpected. To investigate whether mitotic arrest was due to insufficient expression, we co-overexpressed Stg, Cdc2/Cdk1 and Cyclin B in L2

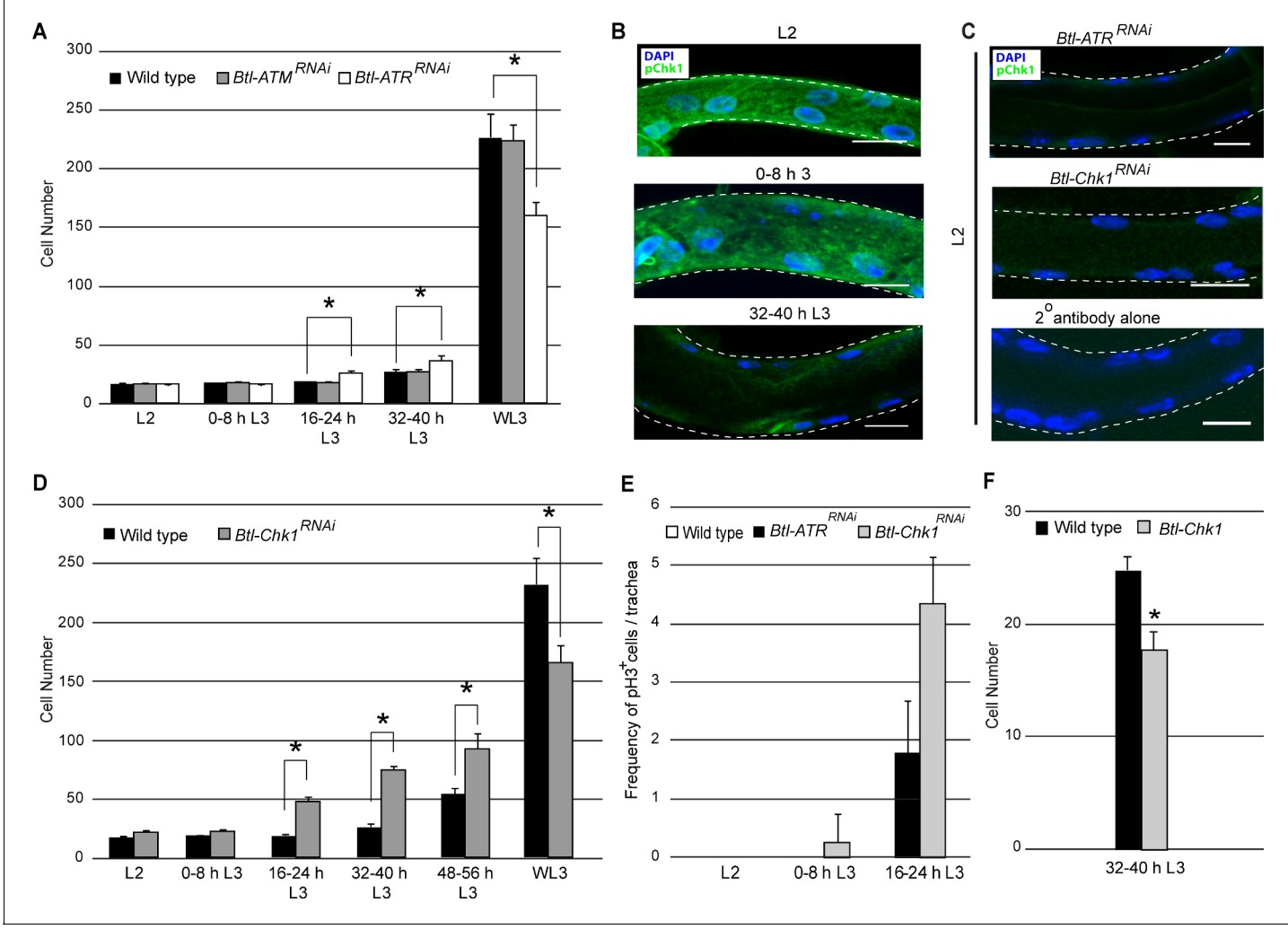

**Figure 2.** ATR and its effector Checkpoint kinase 1 regulate G2 arrest. (**A**) Effect of reduction of *ATR* or *ATM* expression on cell number in Tr2 DT. Graph shows cells numbers in wild type (*Btl-Gal4/+; UAS-nuGFP*), *Btl-ATR*RNAi and *Btl-ATM*RNAi at different stages (mean values ± standard deviation, n = 7 tracheae per condition per timepoint). (**B–C**) Activated Chk1(phospho-Chk1Ser345) immunostaining (green) in Tr2 DT in wild type, *Btl-ATR*RNAi and *Btl-Chk1*RNAi. (**B**) phospho-Chk1Ser345 immunostaining in Tr2 DT in wild type (*Btl-Gal4/+; UAS-nuGFP*) at different stages (**C**) phospho-Chk1Ser345 immunostaining in Tr2 DT in L2 larvae from *Btl-ATR*RNAi (top panel) in *Btl-Chk1*RNAi (middle panel) and wild type treated with secondary antibody alone (bottom panel). (**D**) Effect of reduction in *Chk1* expression in tracheae on cell number in Tr2 DT. Graph shows cells numbers in Tr2 DT in wild type (*Btl-Gal4/+; UAS-nuGFP*) and *Btl-Chk1*RNAi at different stages (mean values ± standard deviation, n = 10 tracheae per condition per timepoint). (**E**) Mitotic indices of Tr2 DT in wild type, *Btl-ATR*RNAi and *Btl-Chk1*RNAi expressing larvae at different stages. Graph shows the frequency of pH3+ nuclei/tracheae at the indicated stages (mean values ± standard deviation, n = 7 tracheae per condition per timepoint). (**F**) Effect of overexpression of *Chk1* on cell number in Tr2 DT. Graph shows cells numbers in Tr2 DT in wild type (*Btl-Gal4/+; UAS-nuGFP*) and *Btl-Chk1*at 32–40 h (mean values ± standard deviation, n = 10 tracheae per condition). DAPI, blue. Scale bar = 10 µm. Student's paired t-test: *p<0.05.

DOI: https://doi.org/10.7554/eLife.29988.007

The following source data and figure supplement are available for figure 2:

**Source data 1.** *Figure 2D*: Numerical data for number of cells in Tr2 DT of wild type and Chk1RNAi -expressing larvae at different stages.
DOI: https://doi.org/10.7554/eLife.29988.009

**Figure supplement 1.** Precocious proliferation in Chk1 mutants is dependent on Stg.
DOI: https://doi.org/10.7554/eLife.29988.008

and early L3 via heat shock and counted cells in Tr2 DT post heat shock. The co-overexpression of Stg, Cdc2/Cdk1 and Cyclin B did not result in increased numbers of cells (*Figure 1K*). This suggestedthat G2 arrest in tracheoblasts could be due to the expression of negative regulators of G2-M and/or the paucity of other positive regulators.

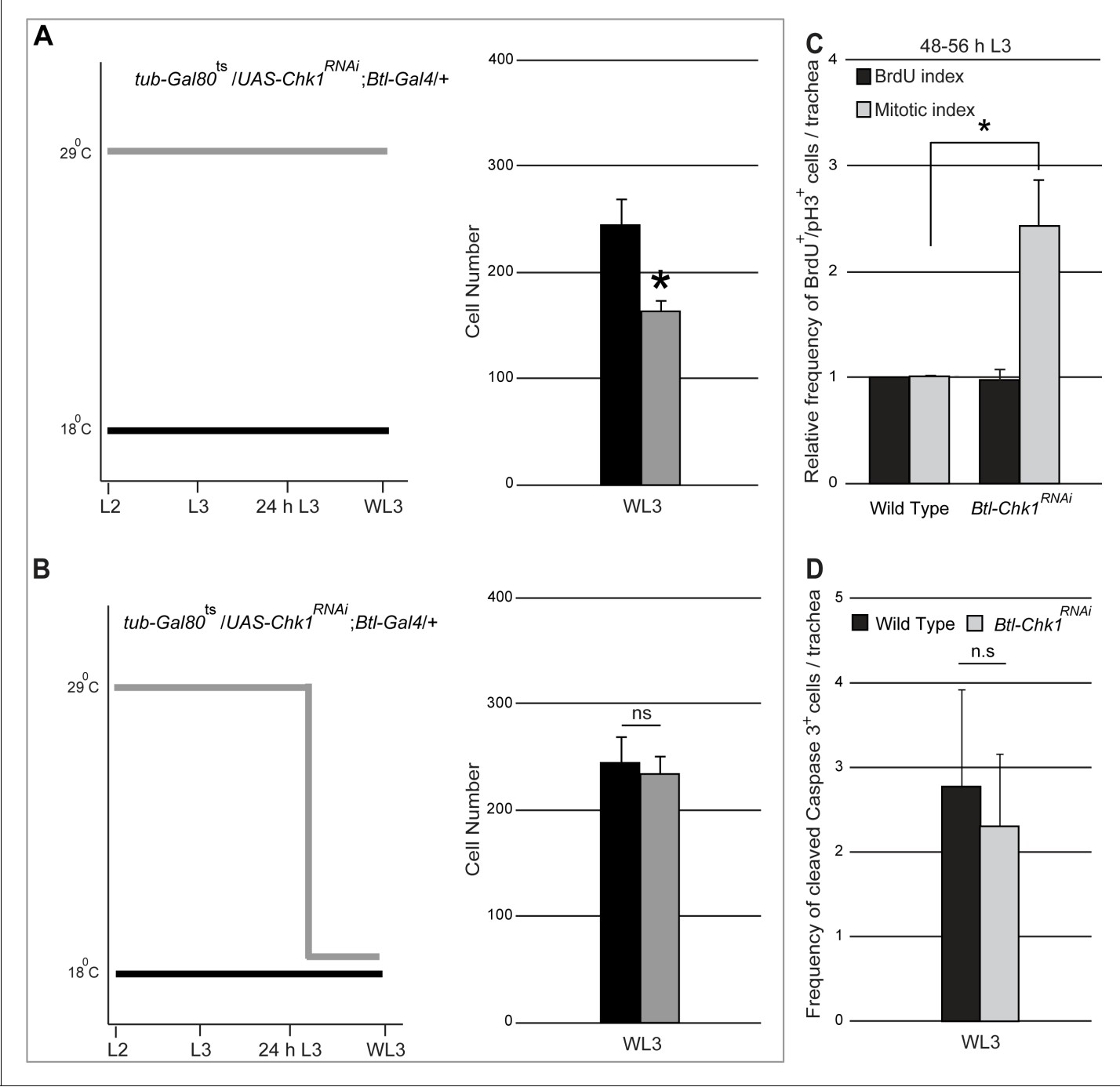

**Figure 3.** Role for Chk1 in the regulation of tracheoblast proliferation post mitotic entry. (**A**). Graph shows cell numbers in Tr2 DT at WL3 in *tub-Gal80ᵗˢ/UAS-Chk1ᴿᴺᴬⁱ; Btl-Gal4/+*larvae grown at 18°C (uninduced (wild type), black) or grown at 29°C from the embryonic period (constitutively induced, grey) (mean ± standard deviation, n = 10 tracheae per condition). (**B**) Graph shows cell numbers in Tr2 DT at WL3 in *tub-Gal80ᵗˢ/UAS-Chk1ᴿᴺᴬⁱ; Btl-Gal4/+*larvae grown at 18°C (uninduced (wild type), black, same as (**A**)) or grown at 29°C from the embryonic period till 24 h L3 then shifted to 18°C till WL3 (conditionally induced, grey) (mean ± standard deviation, n = 10 tracheae per condition). (**C**) Graph showing relative frequencies of BrdU⁺ or pH3⁺ cells in Tr2 DT in wild type (*UAS-nuGFP; Btl-Gal4/TM3,Ser*)and *Btl-Chk1ᴿᴺᴬⁱ* at 48–56 h (mean ± standard deviation, n = 10 tracheae per timepoint). (**D**) Graph showing frequency of cleaved-Caspase 3⁺cells in Tr2 DT in wild type (*UAS-nuGFP; Btl-Gal4/TM3,Ser*) and *Btl-Chk1ᴿᴺᴬⁱ* WL3 larvae (mean ± standard deviation, n = 13 tracheae each). Student's paired t-test: *p<0.05. n.s = not significant.

DOI: https://doi.org/10.7554/eLife.29988.010

As mentioned earlier, downregulation of Cdc2/Cdk1 activity by the ATR-Chk1 or the ATM-Chk2 kinase cascades can lead to arrest in G2. Next we investigated whether ATR and ATM have any role in the regulation of G2 arrest in Tr2. We knocked down ATR or ATM (*Telomere Fusion (Tefu)*, hereafter referred to as ATM) levels in tracheae by RNAi (*Btl-ATR*[RNAi], *Btl-ATM*[RNAi]) and counted the numbers of cells in Tr2 DT at L2, early L3, 32–40 h L3 and WL3. The numbers of cells in Tr2 DT in *Btl-ATR*[RNAi] larvae were significantly higher than wild type by 16–24 h in L3 (21–26 cells compared to 16–18 cells, *Figure 2A*) while there was no increase in cell number in *Btl-ATM*[RNAi] expressing animals. This finding implicated the ATR-Chk1 axis and not the ATM-Chk2 axis in the regulation of G2 arrest in Tr2 DT.

To investigate the role of ATR-Chk1 further, we stained larvae with antisera against an activated (phosphorylated) form of Chk1(*Sørensen et al., 2004*). Anti-phospho-Chk1 (pChk1) immunostaining was high in L2 and early L3 and low at 32–40 h L3 (*Figure 2B*, n = 6–8 tracheae per condition per experiment, n = 3) and subsequent stages (WL3, data not shown). We stained for pChk1 in *Btl-ATR*[RNAi] animals and found no detectable signal indicating that Chk1 phosphorylation is indeed ATR-dependent (*Figure 2C*). To confirm that pChk1 staining was specific, we stained *Btl-Chk1*[RNAi] expressing animals with the same antisera. pChk1 staining in *Btl-Chk1*[RNAi] was comparable to specimens stained with the secondary antibody alone (*Figure 2C*, n = 3 experiments). Next we determined whether the loss of Chk1 also perturbed the timecourse of cell proliferation in Tr2 DT. Cell counts of *Btl-Chk1*[RNAi] larvae showed that the number of cells in in Tr2 DT is higher by 16–24 h in L3 (40–50 cells, *Figure 2D*). In addition to counting numbers of cells, we assayed mitotic activity in *Btl-ATR*[RNAi] and *Btl-Chk1*[RNAi] by anti-pH3 immunostaining at L2, 0–8 h L3 and 16–24 h L3. As indicated in the previous section there are no pH3[+] cells in wild type Tr2 tracheae at these stages (*Figure 1—figure supplement 1*). We observed pH3[+] mitotic figures in *Btl-ATR*[RNAi] at 16–24 h and in *Btl-Chk1*[RNAi] expressing animals at 0–8 h L3 and 16–24 h L3 (*Figure 2E*). Whether the observed differences between *Btl-ATR*[RNAi] and *Btl-Chk1*[RNAi] are due to differences in the efficiencies of the

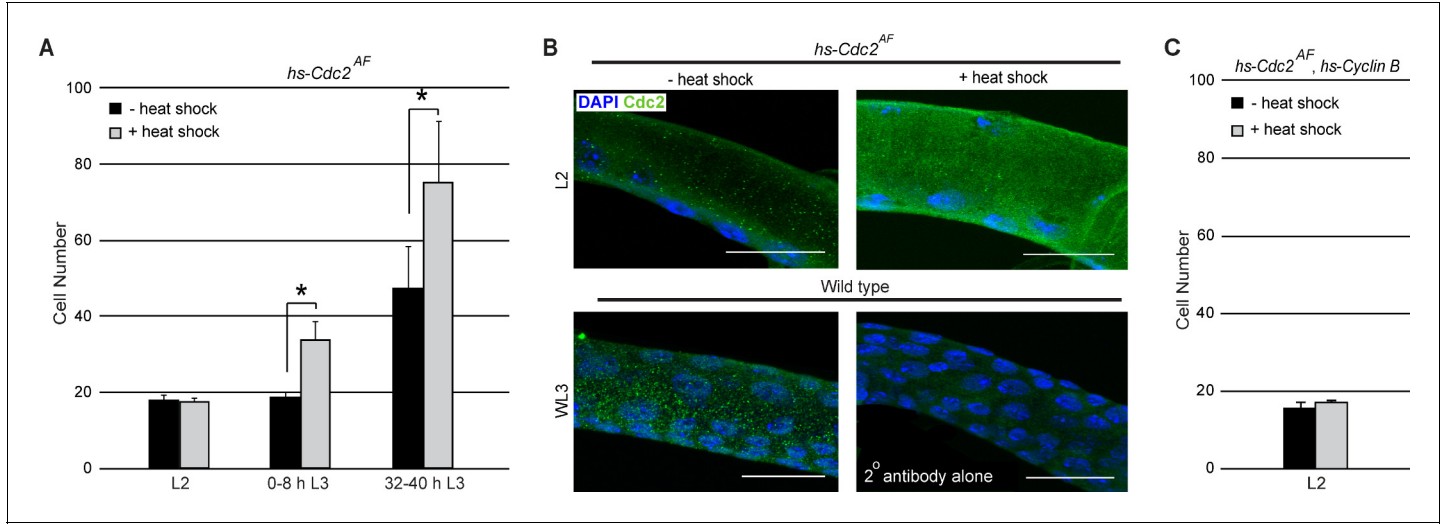

**Figure 4.** Expression of an activated Cdc2 (Cdc2[AF]) is unable to induce mitosis in L2. (**A**) Effect of heat shock-dependent expression of Cdc2[AF] on cell number in Tr2 DT at different stages. Graph shows cell numbers in uninduced (black bars) or induced (grey bars) larvae (mean ± standard deviation, n = 10 tracheae per condition per timepoint). (**B**) Cdc2 immunostaining (green) in Tr2 DT in *hs-Cdc2*[AF] uninduced (top left panel) and induced (top right panel) larvae in L2. Shown in the bottom panels are Cdc2 immunostaining of wild type Tr2 DT at WL3 (bottom left panel) and of wild type Tr2 DT stained with the secondary antibody alone (bottom right panel). (**C**) Effect of heat shock-dependent co-overexpression of *Cdc2*[AF] and *Cyclin B* on cell number in Tr2 DT in L2. Graph shows cell numbers in uninduced (black bars) or heat-shock induced (grey bars) larvae (mean ± standard deviation, n = 5 tracheae per condition per timepoint). Scale bar = 20 µm. Student's paired t-test: *p<0.05.

DOI: https://doi.org/10.7554/eLife.29988.011

The following source data is available for figure 4:

**Source data 1.** *Figure 4A*: Numerical data for number of cells in Tr2 DT of hs-Cdc2[AF] expressing larvae with or without heat shock exposure at different stages.

DOI: https://doi.org/10.7554/eLife.29988.012

respective RNAi lines or due to the presence of other negative regulators of Chk1 is currently unclear. Next we examined how Chk1 overexpression impacted the timing of mitotic entry in Tr2 DT. The overexpression of Chk1 under *Btl* control (*Btl-Chk1*) inhibited mitotic entry at 32–40 h L3 (*Figure 2F*). Based on these findings we conclude that ATR/Chk1 are negative regulators of G2-M in tracheoblasts.

## ATR/Chk1 also regulate rate of cell division post mitotic entry

Despite precocious mitotic entry in ATR/Chk1-deficient animals, the numbers of cells in Tr2 DT at WL3 in these animals were significantly lower than in wild type (*Figure 2A,D*).To investigate whether Chk1 has a role post-mitotic entry, we expressed *Chk1*[RNAi] in the tracheal system in a timed (conditional) manner using the Gal4-UAS-Gal-80[ts] system (*Figure 3*). The expression of *Chk1*[RNAi] from embryonic stages (animals raised at 29°C till WL3 to inactivate Gal-80) led to a reduction in the numbers of cells at WL3 (*Figure 3A*). However, expression of *Chk1*[RNAi] from embryonic stages till 24 h L3 (animals grown at 29°C from embryonic stages to 24 h L3 and shifted to 18°C from 24 h L3 to WL3) did not lead to a reduction in the numbers of cells in Tr2 DT at WL3 (*Figure 3B*). This suggests that in addition to regulating G2-M, Chk1 has a role in tracheoblasts post mitotic entry.

Next we investigated whether reduced numbers of cells in *Btl-Chk1*[RNAi] at WL3 was due to decreased rate of proliferation, increased apoptosis or both. We measured the frequencies of mitotically active cells (BrdU incorporation, anti-pH3 immunostaining) and of apoptotic cells (activated Caspase3 immunostaining) in *Btl-Chk1*[RNAi] and wild type in late L3. The frequencies of BrdU[+] cells were comparable in *Btl-Chk1*[RNAi] and wild type but the frequency of pH3[+] cells was significantly higher in *Btl-Chk1*[RNAi] (*Figure 3C*). The frequencies of activated Caspase3[+] cells were comparable in *Btl-Chk1*[RNAi] and wild type (*Figure 3D*). Together, reduced numbers of cells in *Btl-Chk1*[RNAi] in late L3 and increased frequency of pH3[+] cells in these animals at the same stage suggests that cells lacking Chk1 divide more slowly on account of a prolonged M phase. These data are consistent with a previously described role for ATR-Chk1 in the regulation of the mitotic spindle during cytokinesis (*Gruber et al., 2011*; *Tang et al., 2006*).

## Expression of an activated form of Cdc2/Cdk1 is also insufficient to override G2 arrest in L2

Cells in Tr2 DT enter G2 late in L1/early L2 and remain in G2 till 24–32 h L3. The loss of ATR/Chk1 results in mitotic entry in early L3, ~24–32 h earlier than normal. The reason cells in *Btl-ATR*[RNAi]/*Btl-Chk1*[RNAi] initiated mitoses in L3 but not earlier was unclear and led us to further probe the dependence on Chk1. Chk1 is thought to mediate G2 arrest by inhibition of Cdc2/Cdk1 (*Xiao et al., 2003*; *O'Connell et al., 1997*). The activation of Cdc2/Cdk1 involves dephosphorylation of residues Threonine-14 and Tyrosine-15. Chk1 may prevent activation by inhibiting the phosphatase Stg and/or by stabilizing the kinases Wee/Myt. Next we examined how expression of an 'activated' form of Cdc2/Cdk1 (Cdc2[AF]) in which Threonine-14 and Tyrosine-15 have been mutated to Alanine and Phenylalanine respectively (*Chow et al., 2003*; *Edgar and O'Farrell, 1990*), impacted proliferation. Cdc2[AF]- was induced via heat shock in L2, 0–8 h L3 and 32–40 h L3 and numbers of cells in Tr2 DT were counted. We observed that Cdc2[AF] induction in L2 did not lead to an increase in cell number but induction at 0–8 h L3 and 32–40 h L3 did (*Figure 4A*). We examined levels of Cdc2 expression at these stages, both prior to and post heat shock, using an anti-cdc2 antibody (n = 6 tracheae per condition per experiment, n = 3 experiments). The staining showed that Cdc2 was expressed in L2 and L3 and upregulated post heat shock (*Figure 4B*). We conclude that the induction of Cdc2[AF] is unable to induce proliferation in L2 but is able to at subsequent stages. We also co-overexpressed Cdc2[AF] and Cyclin B in L2 and found that this combination was also unable to induce division in L2 (*Figure 4C*). The inability of Cdc2[AF] to induce mitoses in L2 was consistent with the lack of proliferation in ATR[RNAi]/Chk1[RNAi] at this stage and implicated an ATR-Chk1-independent process in the regulation of G2 arrest in L2 (see Discussion).

## Regulation of ATR/Chk1 in tracheal progenitors

Our analysis of Tr2 DT has shown that Chk1 is phosphorylated in an ATR-dependent manner (*Figure 2*) and that levels of pChk1 are high in L2 and early L3 and diminished at 32–40 hr L3 upon mitotic entry (*Figure 2*). ATR is known to be recruited to double-strand breaks in DNA that result

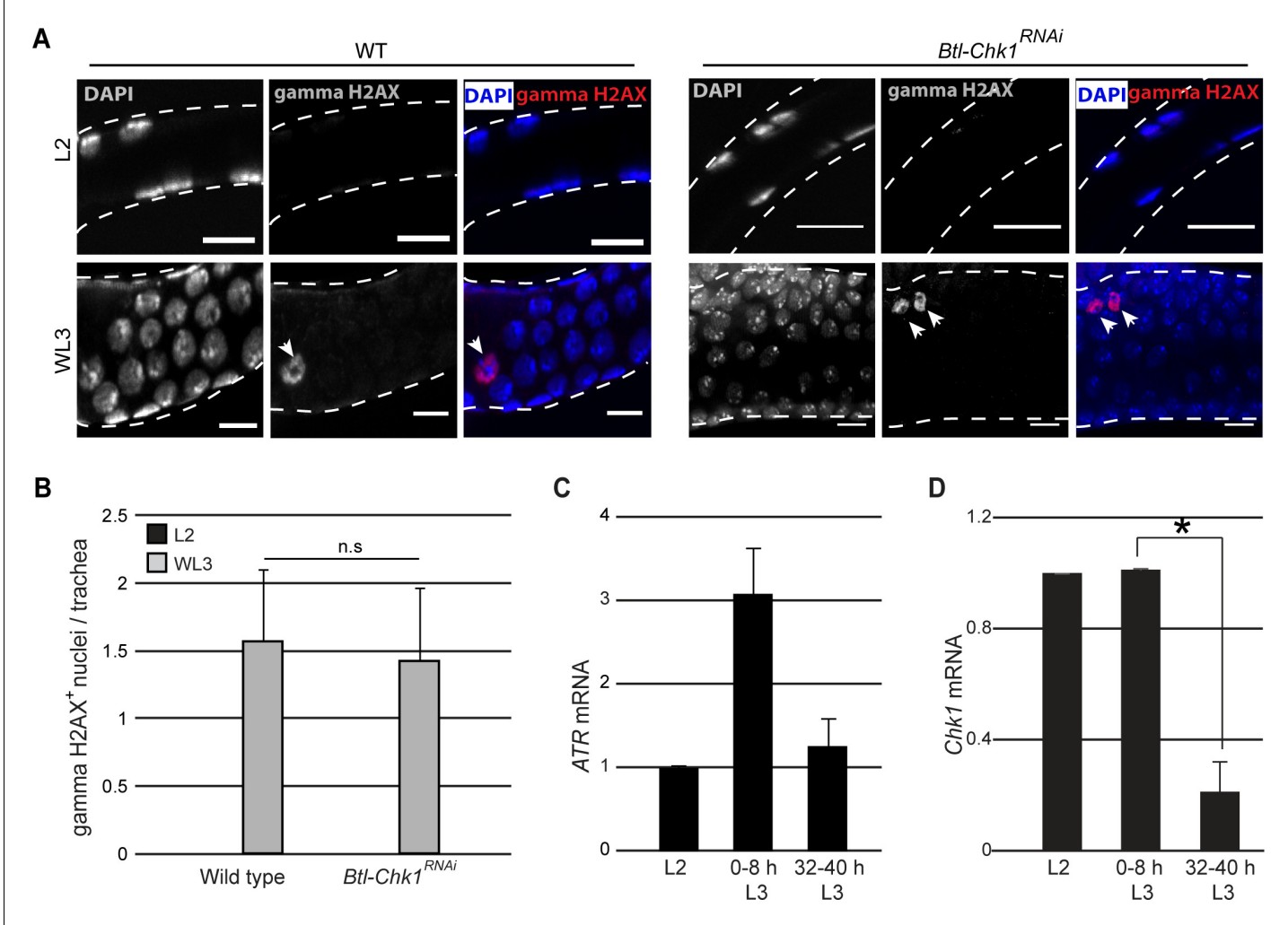

**Figure 5.** Chk1 is regulated at a transcriptional level in Tr2 tracheoblasts. (**A**) gamma-H2AX[Ser139] immunostaining (red, arrowhead) in Tr2 DT in wild type (*Btl-Gal4/+; UAS-nuGFP,* left panels, arrowheads) and *Btl-Chk1[RNAi]* (right panels) at L2 and WL3. (**B**) Frequencies of gamma-H2AX stained nuclei in Tr2 DT of wild type and *Chk1[RNAi]*-expressing larvae at L2 (black bars) or WL3 (grey bars) (mean ± standard deviation, n = 7 tracheae per condition per timepoint).(**C**) Quantitative PCR analysis of *ATR* mRNA levels in Tr2 DT fragments at different stages. Graph shows fold change in mRNA levels with respect to L2 (mean ± standard deviation). (**D**) Quantitative PCR analysis of *Chk1* mRNA levels in Tr2 DT fragments at different stages. Graph shows fold change in mRNA levels with respect to L2 (mean ± standard deviation). Scale bar = 10 µm. Student's paired t-test: *p<0.05.
DOI: https://doi.org/10.7554/eLife.29988.013

from exposure to DNA damaging agents, collapsed/stalled replication forks during DNA synthesis or DNA recombination (*Branzei and Foiani, 2008*). The recruitment of ATR to DNA results in the phosphorylation of the histone variant H2AX (gamma-H2AX/gamma-H2AV in Drosophila) and other proteins including Chk1. Antisera against gamma-H2AX are able to detect DNA damage in Drosophila (*Bayer et al., 2017*; *Rogakou et al., 1999*). Larvae subject to X-ray irradiation-induced DNA damage showed increased levels gamma-H2AX staining in larval tissues (*Bayer et al., 2017*). To investigate whether the activation of ATR/Chk1 in Tr2 DT is associated with double-strand breaks in DNA in these cells, we examined the distribution of gamma-H2AX in L2, early L3 and WL3 (*Figure 4*, n = 6–8 tracheae per condition per experiment, n = 3 experiments). We detected no gamma-H2AX staining in Tr2 DT in L2 and sporadic staining in WL3 (*Figure 5A–B*, early L3 not shown). To validate the gamma-H2AX antisera used in our experiments we stained imaginal discs from larvae irradiated with X-rays at 20 Gy and 40 Gy, 1 h post irrradiation (n = 5–6 imaginal discs per condition per experiment, n = 2 experiments). The frequency of gamma H2AX[+] cells was higher in significantly higher in treated discs than controls and higher in discs exposed to 40 Gy than 20 Gy (data not shown). Thus,

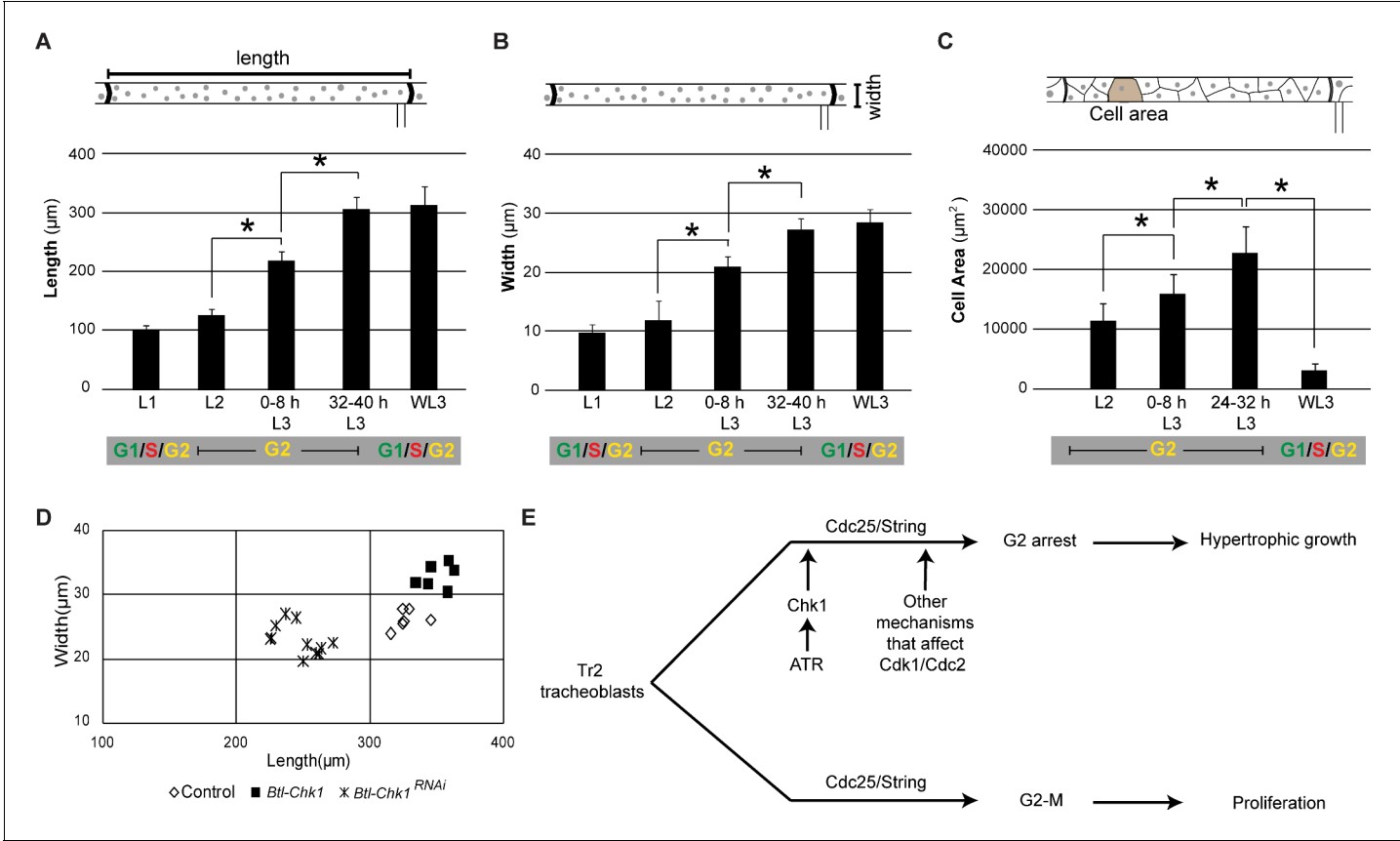

**Figure 6.** Induction of precocious mitoses interferes with growth of Tr2 DT. (**A**) Graph shows length (μm) of Tr2 DT at different stages (mean ± standard deviation, n = 7 tracheae per timepoint). (**B**) Graph shows width of Tr2 DT (μm) at different stages (mean ± standard deviation, n = 7 tracheae per timepoint). (**C**) Graph shows area (μm$^2$) of cells in Tr2 DT at different stages (mean ± standard deviation, n = 15 cells per timepoint). (**D**) Effect of *Chk1*[RNAi] or *Chk1 expression* in size of Tr2 DT at 32–40 h L3. Scatter plot shows length and width of Tr2 DT in tracheae from wild type (n = 6), *Btl-Chk1*[RNAi] (n = 11) and *Chk1*-expressing larvae (n = 7) at 32–40 h L3. (**E**) Model for the regulation of proliferation and growth in Tr2 DT. We propose that negative regulation of G2-M by ATR/Chk1 and by other mechanisms that regulate Cdc2/Cdk1 activity facilitate cellular growth and tracheal hypertrophy. Student's paired t-test: *p<0.05.

DOI: https://doi.org/10.7554/eLife.29988.014

The following source data and figure supplement are available for figure 6:

**Source data 1.** *Figure 6A-C*: Numerical data for measurements of length (6A), width (6B) and cell area in wild type (6C) larvae at different stages.
DOI: https://doi.org/10.7554/eLife.29988.016
**Figure supplement 1.** Cell and tracheal size changes in Tr2 DT.
DOI: https://doi.org/10.7554/eLife.29988.015

we found no evidence for double-strand breaks in DNA in cells paused in G2. We also examined gamma-H2AX levels in *Btl-Chk1*[RNAi] animals to see if the levels were elevated. Here again, we detected no gamma-H2AX in L2 and some labeling, at levels comparable to wild type, at WL3 (*Figure 5A–B*). The sporadic gamma-H2AX staining observed in wild type and *Btl-Chk1*[RNAi] animals is intriguing and the underlying reason is currently unclear. It has been reported that the Ser-139 residue on H2AX is also phosphorylated in cells undergoing apoptosis (*Rogakou et al., 2000*). Since the frequencies of gamma-H2AX[+] nuclei/trachea and activated-Caspase3[+] cells/trachea in control and Btl-Chk1[RNAi] at WL3 are comparable (compare *Figure 5B* and *Figure 3D* ), it is plausible that some of the gamma-H2AX[+] nuclei in control and Btl-Chk1[RNAi] tracheae at WL3 are apoptotic cells.

In an independent set of experiments, we probed the expression of ATR/Chk1 mRNA in Tr2 DT at different larval stages (L2, 0–8 h L3, and 32–40 h L3) by qPCR (≥15 tracheal fragments per timepoint per experiment, n = 3 experiments). With respect to expression at L2, levels of ATR expression were slightly increased at 0–8 h L3 (*Figure 5C*). Interestingly, the levels of Chk1 mRNA were ~5 fold higher at both L2 and 0–8 h L3 in comparison to 32–40 h L3 (*Figure 5D*). This suggests that the high

levels of pChk1 in Tr2 DT in L2 and early L3 could be regulated at a transcriptional level via the regulation of Chk1 expression.

## G2 arrest facilitates growth

The larval tracheal system consists of a network of epithelial tubes that grow in length and in circumference as the larva grows. As indicated earlier (*Figure 1*), Tr2 DT and the cells that comprise this segment grow in size during larval stages. Measurements of the length and width of Tr2 DT at different larval stages showed that most of the growth of this segment occurs from the time the animals enter L2 till mid-L3 (*Figure 6A–B*). During this period, Tr2 DT grows in length by 268 ± 32% (n = 6 tracheae) and in width by 247 ± 19% (n = 12 tracheae). We also estimated the sizes of tracheoblasts in Tr2 DT (2D area) over the L2-L3 interval. For this, tracheae were stained with Phalloidin to delineate margins of cells and cellular areas were measured (see *Figure 6—figure supplement 1*). Between L2 and 24–32 h L3, cells virtually doubled in size (cellular area increased by 112 ± 37% (n = 15)) (*Figure 6C*). Thus, analysis of the growth trajectory of Tr2 DT, and the tracheoblasts that comprise it,showed that maximal growth occurs in the period when the tracheoblasts are paused in G2. This raised the possibility that the arrest in G2 facilitates the growth of cells and the tracheal branches they comprise.

Next we investigated how abrogating (*Figure 2D*) or prolonging G2 arrest (*Figure 2F*) impacted growth of Tr2 DT. Measurements of length and width of Tr2 DT in *Btl- Chk1^{RNAi}* animals at 32–40 hr L3 showed that the growth was significantly reduced in comparison to wild type (*Figure 6D*, both average length and width are statistically different p<0.05, see Source data file 1). Conversely, the measurements of length and width of Tr2 DT in animals overexpressing Chk1 showed thatTr2 DT segments were significantly larger in this background (*Figure 6D*, p<0.05, both average length and width are statistically different p<0.05, see Source data file 1). We then compared the sizes of trachea in wild type, *Btl- Chk1^{RNAi}* and Btl-*Chk1* animals at 0–8 h L3 and found that they are comparable (*Figure 6—figure supplement 1*). Based on these findings, we conclude that ATR/Chk1-dependent G2 arrest facilitates cellular and organ growth in Tr2.

## Discussion

Here we investigate the mechanism for and implication of G2 arrest in progenitors of adult thoracic tracheal system in Drosophila. We show that Tr2 tracheoblasts remain arrested in G2 for ~48–56 h during larval life during which the cells and the tracheae they comprise grow in size. Our findings are that tracheoblasts paused in G2 express both the essential drivers for G2-M like Cdc2/Cdk1, Cyclin B and Stg, and negative regulators of G2-M like ATR and Chk1, and that the G2-M transition in these cells involves the coordination of several genetically distinguishable processes including the downregulation of Chk1. Our analysis also reveals that arrest in G2 is necessary for growth of tracheoblasts and the tracheae they comprise. In the sections that follow we discuss the processes underlying G2 arrest, the relationship between G2 arrest and cellular growth and the broader implications of the developmental program described here.

ATR/Chk1 have been implicated in the negative regulation of Cdc2/Cdk1 activity leading to a slowdown of the S/G2 phases of the cell cycle (*Branzei and Foiani, 2008*; *Su et al., 1999*; *Blythe and Wieschaus, 2015*). In these contexts, ATR is recruited to stalled/collapsed replication forks during DNA replication and to double-strand DNA breaks that occur upon exposure DNA damaging agents or during recombination (*Kumagai et al., 2004*; *Blythe and Wieschaus, 2015*). Recruitment of ATR leads in turn to the phosphorylation and activation of Chk1 and to the inhibition of Cdc2/Cdk1 activity. The inhibition of ATR/Chk1 has been shown to hinder completion of DNA replication and DNA repair and result in aberrant mitoses; ATR/Chk1 mutant animals are embryonic lethal (*Liu et al., 2000*; *Sibon et al., 1999*). In the context of tracheoblasts, we find that ATR/Chk1 act to arrest cells in G2. But unlike in other contexts, we find no evidence for any DNA damage in cells in which Chk1 is active nor any increase in DNA damage upon reduction of Chk1 levels and the induction of precocious mitoses. We independently measured the frequency of apoptotic nuclei in Chk1 mutants to find that it is indistinguishable from wild type. Thus, it appears unlikely that the activation of ATR/Chk1 in Tr2 DT is the outcome of the activation of the canonical DNA damage checkpoint. The mechanisms for the activation of ATR/Chk1 in Tr2 DT merit further investigation. Analysis of ATR/Chk1 expression by qPCR suggests that levels of Chk1 mRNA are significantly higher in

arrested cells. Increased expression of Chk1 mRNA could contribute toward increased levels of phosphorylated Chk1 and to G2 arrest.

Activated Chk1 is thought inhibit Cdc2/Cdk1 by either inhibiting Stg and/or stabilizing Wee/Myt (*O'Connell et al., 1997*; *Xiao et al., 2003*). We find that the expression of Wee$^{RNAi}$ and Myt1$^{RNAi}$ does not recapitulate the Chk1$^{RNAi}$ phenotype (data not shown). This suggests that Wee/Myt1 do not contribute toward G2 arrest in tracheoblasts and that Chk1 acts via the inhibition of Stg. We have examined if the precocious proliferation in *Btl-Chk1*$^{RNAi}$ animals is dependent on Stg. Counts of cell numbers in Tr2 DT in *Btl-Chk1*$^{RNAi}$*Stg*$^{RNAi}$ animals showed that there was no increase at 16–24 h L3 or at 32–40 h L3 (*Figure 2—figure supplement 1*). This is consistent with the possibility that Chk1 inhibits mitotic activity by inhibiting Stg.

Tracheoblasts lacking ATR/Chk1 pause in G2 for 24 h (L2) and proliferate ~24–32 h prior to the normal time for mitotic entry (early L3). We find that expression of a constitutively active form of Cdc2/Cdk1, that is insensitive to ATR/Chk1, is unable to induce precocious mitotic exit in cells in L2. This shows that the mechanism for arrest in L2 is likely to be independent of ATR/Chk1. Mitochondrial fragmentation that occurs in G2 is necessary for the segregation of these organelles into daughter cells during mitosis. Several studies have shown that the inhibition of mitochondrial fission in actively proliferating cells leads to arrest in G2 (*Westrate et al., 2014*; *Lee et al., 2014*). Importantly, a recent study has shown that cells in which mitochondrial fission was inhibited were also unresponsive to expression of Cdc2$^{AF}$(*Lee et al., 2014*). Whether the timecourse of mitochondrial fission in Tr2 DT impacts mitotic competence remains to be determined.

Arrest in G2 correlates with increase in size of tracheoblasts and hypertrophic growth of the tracheal branch they comprise (DT). Based on the quantitation of tracheal size (*Figure 6*), we estimate that the tracheal epithelium, and at least some of the tracheoblasts that comprise it, grow in volume ~13 fold in the L2-32-40 h L3 interval. The association between G2 arrest and enormous cellular growth has been reported in other developmental contexts. Histoblasts arrested in G2 grow ~60 fold in volume (*Ninov et al., 2009*) and Drosophila spermatocytes arrested in pre-meiotic G2 arrest grow ~25 fold in volume (*Ueishi et al., 2009*). We find that abrogating G2 arrest in developing tracheoblasts via knockdown of ATR/Chk1 diminishes cellular and organ growth. Conversely, prolonging G2 arrest leads to increased growth. It is plausible that arrest in G2 facilitates cellular growth. This cellular growth may drive hypertrophic organ growth as we show here and facilitate rapid proliferation subsequently.

Analysis of the mechanisms underlying hypertrophic growth in the vertebrate kidney (*Shankland and Wolf, 2000*) and heart (*Braun-Dullaeus et al., 1999*) has shown that it is associated with cells in G1 arrest. The findings presented here suggest that G2-arrested cells may also be relevant to hypertrophic organ growth and pathogenesis. Interestingly, hypertrophic cellular growth in G1-arrested cells referred to above has been shown to be dependent on the juxtaposition of cell cycle activators and inhibitors. In the context of glomerular cell hypertrophy associated with diabetic nephropathy, elevated glucose levels induced quiescent cells to enter G1 while Angiotensin II and TGF-Beta signaling elevated levels of CDK inhibitors that prevented entry into S phase (*Fujita et al., 2004*). The negative regulation of cell cycle progression in cells that are mitotically active may be necessary for the hypertrophic growth of G2-arrested cells as well.

## Materials and methods

### Fly strains and handling

The following strains were obtained from repositories: *UAS FUCCI*, *TubGAL80*$^{ts}$;*TM2/TM6b,Tb*, *UAS-ATM*$^{RNAi}$ (Bloomington Drosophila Stock Center), *UAS-Chk1*$^{RNAi}$, *UAS-Stg*$^{RNAi}$, and *UAS-ATR*$^{RNAi}$(Vienna Drosophila Resource Center), *UAS-Chk1* (In-house fly facility). The following strains were received as gifts: *Btl-Gal4, hs-String, hs-Cdc2, hs-Cdc2*$^{AF}$, *hs-CyclinB*. Strains were raised on a diet of cornmeal-agar and maintained at 25°C except *hs* and GAL80$^{ts}$ strains that were maintained at 22°C and 18°C respectively. For experiments involving GAL80$^{ts}$ strains, the animals were moved to 29°C at indicated stages for indicated time periods. For experiments involving *hs* strains, the animals were heat-shocked at 37°C for 30 min at indicated stages, transferred to 22°C for 3 h and then sacrificed.

## Larval staging

Larval staging was based on the morphology of the anterior spiracles as previously described (*Guha and Kornberg, 2005*). For timepoints in L3, L2 larvae were collected and examined at 8 h intervals to identify animals that had undergone the L2-L3 molt (0–8 h L3). 0–8 h L3 cohorts isolated in this manner were staged for subsequent timepoints.

## BrdU incorporation

Larvae were fed 5-Bromo-2'-deoxyuridine (1 mg/ml final concentration, Sigma, in cornmeal-agar) for 2 hr and sacrificed for analysis.

## Immunohistochemistry, imaging and morphometry

Animals were dissected in PBS and fixed with 4% (wt/vol) Paraformaldehyde in PBS for 30 min. Immunohistochemical analysis utilized the following antisera: Rabbit anti-gamma-H2AX (Novus Biologics, 1:300), Rabbit anti-cleaved-Caspase3 (Cell signaling technology, 1:300),Rabbit anti-pH3 (Millipore, 1:500), Guinea pig anti-stg (gift from Dr. Yukiko Yamashita, 1:500 [*Inaba et al., 2011*]), Rabbit anti-Cdk1/Cdc2 (PSTAIR) (Millipore, 1:500), Mouse anti-CyclinB (DSHB, 1:300), Rabbit anti-phospho Chk1 (Abcam, 1:200), Mouse anti-BrdU (Sigma, 1:100) and Alexa 488/568/647-conjugated Donkey/ Goat anti-mouse/rabbit/guinea pig secondary antibodies (Invitrogen, 1:300). Tyramide signal amplification was used as per manufacturer recommendations for pChk1 detection. As part of this protocol the following reagents were used: Tyramide amplification buffer and Tyramide reagent (Invitrogen), Vectastain A and B and Biotinylated goat anti Rabbit IgG (1:200, Vector Labs). Tracheal preparations were flat-mounted in ProLong Diamond AntifadeMountant with DAPI (Molecular Probes)and imaged on Leica TCS SP5 or Zeiss LSM-780 laser-scanning confocal microscopes. Images were processed using Image J and Adobe Photoshop. For quantification of cell number, fixed specimens were mounted in ProLong Diamond AntifadeMountant with DAPI and the number of nuclei were counted on a Zeiss Axio Scope A1 microscope.

The DT of the second thoracic metamere was identified morphologically based on the cuticular banding pattern at anterior and posterior junctions. Length of DT was measured as the distance between the cuticular bands of respective metameres. Area of a cell was measured by creating a mask over the cell boundaries as stained by Alexa 568 Phalloidin (Molecular Probes, 1:500) using ImageJ.

## DNA quantitation

*Btl*-FUCCI-expressing larvae were dissected in PBS, incubated with 5 µM Draq5 (Abcam) for 5 min (PBS), fixed in 5% paraformaldehyde (PBS) for 10 min, and rinsed in PBS + 0.3% Triton X-100. Trachea were flat-mounted in 50% (vol/vol) glycerol and compressed gently to expel air from the tubes to eliminate light scattering by air-filled tracheal tubes. Nuclear regions were selected using ImageJ and intensity of DRAQ5 fluorescence was estimated.Intensities were corrected for background by subtracting an average intensity value of five selected regions devoid of nuclei from all pixels in the image.

## RNA isolation and quantitative PCR

The DT of the second thoracic segment was identified morphologically as described in the previous section, micro-dissected with forceps in PBS and transferred to Trizol reagent (Ambion) on ice for RNA extraction as per instructions provided by the manufacturer. RNA was then precipitated using Isopropanol/4M Lithium chloride. cDNA was synthesized (100 ng RNA) using Maxima First Strand cDNA Synthesis Kit (Thermo Fisher). 1 µL of cDNA was then used to perform qPCR using the SYBR Green (Maxima Probe/ROX qPCR Master Mix, Thermo Fisher) protocol. Primer sequences for candidate genes and *GAPDH* (internal control) are provided below. Relative mRNA levels were quantified using the formula RE = $2^{-\Delta\Delta Ct}$ method.

The following primer sets were used:

| | |
|---|---|
| GAPDH forward | 5' CGTTCATGCCACCACCGCTA 3' |
| *GAPDH* Reverse | 5' CACGTCCATCACGCCACAA 3' |

*Continued on next page*

| *String* Forward | 5' CAGCATGGATTGCAATATCAGTAAT 3' |
| *String* Reverse | 5' AGACCCATCAGCTCCGGACT 3' |
| *Chk1* Forward | 5' AACAACAGTAAAACGCGCTGG 3' |
| *Chk1* Reverse | 5' TGCATATCTTTCGGCAGCTC 3' |
| *ATR* Forward | 5' CCAGATAGCAGCGAGTGCAT 3' |
| *ATR* Reverse | 5' CGAGGTCCAGGGAACTTAGC 3' |
| *Cdc2* Forward | 5' CCATCAACCGCGATCAGAGAAAT 3' |
| *Cdc2* Reverse | 5' CTCTCCATGTGCTTATCAACTGGC 3' |
| *Cyclin B* Forward | 5' TTACAGGCCATCGGAGATTGC 3' |
| *Cyclin B* Reverse | 5' TTCGCGATCAGCCGGGTAAT 3' |

## Acknowledgements

We thank Bruce Edgar, Norbert Perrimon, Christian Lehner, Shigeo Hayashi, Nikita Yakubovich and Patrick O'Farrell for fly strains, Yukiko Yamashita for providing us with Cdc25/String antibody, the Central Imaging and Flow Cytometry Facility (CIFF) at inStem and Fly Facility at C-CAMP for their support, Priyasha Mishra for technical assistance, Volker Hartenstein for sharing images of the larval tracheal system, and Apurva Sarin, Tina Mukherjee,Sunil Laxman and Narmada Khare for critical reading of the manuscript. Support: Ramalingaswami Fellowship (Department of Biotechnology, Government of India, AG) and institutional funds from inStem (AK, AB).

## Additional information

### Funding

| Funder | Grant reference number | Author |
|---|---|---|
| Department of Biotechnology, Ministry of Science and Technology | inStem Core Funds | Amrutha Kizhedathu<br>Arjun Guha<br>Archit V Bagul |
| Ministry of Science and Technology | Ramalingaswamy Fellowship, inStem/DBT/8241 | Arjun Guha |

The funders had no role in study design, data collection and interpretation, or the decision to submit the work for publication.

### Author contributions

Amrutha Kizhedathu, Archit V Bagul, Conceptualization, Formal analysis, Investigation, Visualization, Methodology, Writing—original draft, Writing—review and editing; Arjun Guha, Conceptualization, Resources, Formal analysis, Supervision, Funding acquisition, Writing—original draft, Writing—review and editing

### Author ORCIDs

Arjun Guha http://orcid.org/0000-0002-3753-1484

### Decision letter and Author response

Decision letter https://doi.org/10.7554/eLife.29988.021
Author response https://doi.org/10.7554/eLife.29988.022

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
