## [Decision Letter]

Thank you for sending your article entitled "Negative regulation of G2-M in the presence of Cdc25/String facilitates tracheoblast growth and tracheal hypertrophy in *Drosophila*" for peer review at *eLife*. Your article has been favorably evaluated by Didier Stainier (Senior Editor) and three reviewers, one of whom is a member of our Board of Reviewing Editors.

The reviewers have discussed the paper amongst themselves and come to the conclusion that the paper has merit but is not publishable in its current form, however it could be with very substantial revisions.

Overall, the reviewers were convinced that the ATR-Chk1 checkpoint controls the cell cycle during tracheal development, and felt that this was an interesting use of the checkpoint in development that warrants reporting in *eLife*. However the reviewers were not as convinced by the data on mitochondrial regulation of the cell cycle, or growth control by Stg, and felt that more data would be required to support these aspects of the paper. Thus we'd like to suggest two options you might follow in revising the paper.

First, we think you could remove the data on mitochondria and PINK1, and also the data on Stg and growth, and submit a shorter, simpler paper including only the data relevant to the ATR-Chk1 mechanism, which the reviewers felt was relatively strong. This is your simplest option and could probably be achieved in a reasonable timeframe.

Second, an alternative would be to provide a revision with more data from new experiments addressing mitochondrial fission, PINK1, and Stg-mediated growth. In this case the reviewers would have to be convinced by the new data, and its relevance to the ATR-Chk1 checkpoint. I expect this will be significantly more complex and time-consuming than option #1.

*Reviewer #1:*

In this paper, Guha et al. report a study of cell cycle control during tracheal growth in the *Drosophila* larva. The authors have made several interesting observations. First, tracheoblasts undergo prolonged arrest in G2, and this is mediated in part by inhibition of M-CDK by the well-known ATR-Chk1 system, normally used in response to DNA damage, or incomplete replication. They've also identified an inhibitory role for mitochondria, which need to undergo fission for efficient mitosis, but remain fused during the G2 arrest, apparently inhibiting mitotic progression. The authors provide a careful analysis of cell cycle phenotypes and use a number of excellent genetic tests to demonstrate how the G2 arrest is mediated by ATR/Chk1 and mitochondrial fission. Although there are a number of technical weaknesses that should be addressed (see details below), the demonstration of G2 arrest and its regulation by ATR/Chk1 is convincing enough. The interesting aspect for a general audience is that ATR/Chk1 are used here as developmental regulators rather than in DNA damage response, and there are not very many examples of this general mechanism. The paper stops short of determining how ATR/Chk1 and mitochondrial fission are regulated developmentally. This would of course be nice to know, but isn't essential to make this study valuable. At the end of their paper the authors show some data (Figure 5-6) suggesting G2 arrest promotes growth, and that stg/Cdc25 promotes ribosome biogenesis and interphase growth of tracheoblasts, but this is very preliminary, not directly relevant to the paper's central message. I suggest that this data be removed, and the work on ATR/Chk1 presented a short report.

1) Although the authors' analysis of the tracheoblast cell cycle is careful, they have not measured DNA. As a 4N DNA content is the best marker for G2 arrest, this should be done. The FUCCI markers could be misleading if the arrest mechanism is unusual (subsection “Thoracic tracheoblasts arrested in G2 express Cdc25/Stg and other mediators of G2-M”, second paragraph).

2) Figure 1. Showing a schematic summary of the FUCCI data here, without showing real data, is not acceptable. I suggest that some part of the actual FUCCI data be included in the main figure.

3) The authors argue that stg is expressed during the G2 arrest, and imply that the levels should be sufficient to trigger M phase. However the data don't really support this. Stg levels clearly rise as the cell begin to divide, and the lower levels may be sub-threshold. There is not data in the paper to rule this out, so I suggest the authors moderate their statements that Stg is present during G2 arrest. These statements, in the title, Abstract, and elsewhere are misleading.

4) Figure 2. Tp bolster their analysis and cement their conclusions, the authors should quantify numbers of PH3 figures at timpanist in the WT and Chk1-RNAi and/or ATR-RNAi animals. The one Ph3 figure shown is not even obviously in a dividing cell, and the reliance on cell number counts raises some issues as cell death also occurs in this organ at this time.

5) Figure 3 shows some very raw data that is not obviously conclusive. I suggest the authors design a better experiment to test their idea and present that instead. As is, this figure doesn't add much to the paper. Clonal analysis might be useful here.

6) Better data are required to document the onset of mitochondrial fission in L3. Ideally this data will be quantitative. They should also provide data showing that PINK1 promotes mitochondrial fission in their system.

7) The additional mitoses generated by promoting mitochondrial fission by PINK1 are a relatively few. The authors make a lot of this small effect, but it's not clear that the combination of Chk1 loss and PINK1 completely relieves G2 arrest. From Figure 4F it looks like it does not. Perhaps the induction of Stg in L3 is important, or some other mechanism. The authors should take this into account in their Discussion.

8) The section on growth in G2 is rather descriptive, and the effects of PINK1 and Chk1-RNAi are small and perhaps insignificant. Looking at this data, I would conclude that PINK1 may inhibit growth, not that G2 arrest promotes it. I suggest removing this inconclusive data (Figure 5).

9) As noted above, I suggest removing the data shown in Figure 6, since it is very preliminary and not directly relevant to the paper's central message.

*Reviewer #2:*

The authors investigated how tracheoblast proliferation is regulated during a stage of development when cells become arrested in G2 phase and grow rapidly before re-entering a proliferative phase shortly before pupariation. Unexpectedly the mitotic activator Cdc25 (Stg) as well as activated Chk1 (a negative regulator of mitosis), were both present in these G2 phase arrested cells. Tissue-specific RNA interference against known G2/M regulators indicated that ATR/Chk1 activity was at least partially responsible for maintaining the G2 arrest during L3. This seems plausible since Chk1 is known to negatively regulate Cdc25 and positively regulate Cdk1 inhibitory kinases to prevent mitosis in undamaged cells.

Next they investigated why cells were refractory to induction of precocious mitosis during L2. Expression of Cdk1AF and Cyclin B had no effect in L2 but the cells became susceptible in early L3. Since previous work in other labs showed that interfering with mitochondrial fission could block cell cycle progression they used a reporter to study the behavior of this organelle during tracheal development. They observed a transition in mitochondrial morphology from fused to fragmented preceding the onset of cell proliferation later in L3. To examine how this mitochondrial behavior affected the cell cycle they induced mitochondrial fission prematurely by expressing the PINK1 kinase and observed that susceptibility to premature mitosis by expression of Cdk1AF or Chk1 siRNA was also advanced. This also resulted in a substantial decline in growth of the trachea. These findings suggested that a 'mitochondrial' checkpoint may be operating during earlier developmental stages of the G2 phase arrest to maintain cells in a state compatible with the onset of rapid growth of the trachea. After knocking down Stg to prevent mitosis in late L3 they observed a similar growth reduction and correlated it with a decrease in the nucleolar protein synthesis marker Fibrillarin. In contrast, such treatments had no effect on an adjacent population of tracheoblasts that are endoreplicating. Collectively, these observations led them to the provocative conclusion that Cdc25 is required for growth in this particular developmental context. Such a mechanism for coordinating growth and proliferation has not previously been described to my knowledge.

The manuscript was clearly written and the experiments appear to have been carefully controlled and quantified. Many of the figures were difficult to view however (horizontal orientation).

*Reviewer #3:*

The manuscript by Guha and colleagues examines G2-M regulation in the Tr2 metamere of the *Drosophila* trachea. Interestingly, cells in this region of the trachea have been previously shown to be both functional as tracheal cells in the larvae and then to act as adult progenitors. Here, Guha et al. explore cell cycle regulation in this distinctive group of tracheal progenitors.

Using FUCCI, they define a prolonged G2 in TR2 cells. Using assays for protein and mRNA expression along with genetics, they convincingly show that ATR/Chk1 and Stg are both present during the prolonged G2, and that Chk1 represses premature mitotic entry. Also convincing are the data supporting cell growth during prolonged G2.

Less convincing is the authors' argument that mitochondrial dynamics are important for G2 arrest. The authors present images of mitoGFP as evidence of developmental changes in mitochondrial dynamics, which they argue are altered when Cdc2 activity is manipulated. I find these data unconvincing and can't tell what is being assayed, and there is no quantitation of such data to speak to reproducibility. EM may be needed to more directly assay mitochondrial dynamics. Further, overexpression of a single mitochondrial regulator (pink1) is presented as additional evidence of the role of mitochondria in cell cycle regulation. In addition to being an ectopic gene expression experiment, the effects are relatively minor, and it's unclear whether any effects on cell cycle/cell size are in any way directly altering cell cycle regulation. Messing with mitochondria may disrupt cell fate, which could cause premature cell cycle entry, for example. Additional genetic evidence of a specific role for mitochondrial dynamics is definitely needed.

I was also confused by the author's interpretation of the loss of cells in Chk1 RNAi animals at wandering L3, which they attribute entirely to long mitosis, as opposed to premature mitotic entry. First of all, they directly contradict this interpretation on line 243, where they say that Chk1 loss leads to earlier than normal mitotic entry. Second, longer mitosis on its own should not lead to drastic changes in cell number, as mitosis is usually a very small fraction of the cell cycle. Perhaps in the absence of chk1 there are aberrant mitoses, which lead to cell elimination by a non-apoptotic means, as is widely reported in numerous systems? This possibility is not discussed.

[Editors' note: further revisions were requested prior to acceptance, as described below.]

Thank you for resubmitting your work entitled "Negative regulation of G2-M by ATR (mei-41)/Chk1(Grapes) facilitates tracheoblast growth and tracheal hypertrophy in *Drosophila*" for further consideration at *eLife*. Your revised article has been favorably evaluated by Didier Stainier (Senior Editor) and three reviewers, one of whom is a member of our Board of Reviewing Editors.

The manuscript has been improved a great deal, and the reviewers found the simplified format much superior. However, the reviewers have raised additional points, mostly about presentation, that need to be addressed before acceptance. These are outlined below in the full reviews. We expect that these issues should be fairly easy to address, and look forward to a further revision.

*Reviewer #1:*

The revision of this paper follows the plan agreed upon after the first review, namely to simplify the paper by deleting some weak data and focus the presentation on the novel, interesting cell cycle effects that were discovered during tracheal development. The paper is now concise and largely convincing in its conclusions. However it still has six figures, and I suggest that some of these could be combined, since they are small. The quality of the data has also been improved with some additions as suggested in the reviews. I think the paper is essentially appropriate for publication in this form. However upon reading it, I did note several instances where the data, or the presentation thereof, could be improved. I invite the authors to address these specific points with a further round of minor revisions.

1) For Figure 1, it would be very helpful to provide a graph of PH3+ cell counts at the different stages shown with the other cell cycle markers.

2) Also for Figure 1, the graph showing DNA contents (Figure 1G) is rather confusing. I suggest graphing this data as in Figure 1I, with stage on the X axis and% cells with different ploidies on the Y axis.

3) For Figure 2, we should see an increase in PH3+ cells in the Btl>ATR-RNAi and Btl>Chk1-RNAi samples. Please count the numbers of PH3+ cells for these genotypes and provide graphs. The data as is is substandard.

4) For Figure 6D, the authors should add p values for the growth increases and decreases. Pictures of the larger and smaller trachea would also be nice (but not essential.

5) Regarding Figure 6 and the statement at the beginning of the Discussion, I believe the Btl>Chk1-RNAi trachea have fewer cells due to effects earlier in development (noted earlier in the paper). This reduction in cell numbers might explain their smaller size. The authors should consider this problem with their interpretation and provide additional data using the Btl-TS driver if necessary. If no new data is provided the conclusion may have to be changed.

6) Please provide a citation for the statement in the last paragraph of the subsection “Fused, not fragmented, mitochondria contribute to G2 arrest in tracheoblasts”.

*Reviewer #2:*

Temporal coupling of the G2/M transition with terminal cell fate determination has been well-studied in situations where inhibitory phosphorylation of Cdk1 and lack of Cdc25 (Stg) expression ensure G2 phase arrest. Circumstances where G2 phase arrested cells undergo rapid cell growth followed by a burst of proliferation are less well understood. Such a system was studied here, the Tr2 tracheal system, where Cdc25(Stg), Cdk1 and mitotic Cyclin B were present throughout larval development yet cells remained arrested in G2 phase. RNA interference experiments showed that knockdown of the Chk1 (Grp) checkpoint kinase or to a lesser extent ATR (Mei-41) resulted in precocious mitosis in these cells during early L3 phase shortly before they would normally undergo a hyperproliferative state. These data support a developmental role for Chk1 (and ATR) in maintaining G2 phase. Although the mitotic index was elevated under these RNAi conditions, cell counts in late L3 indicated that the numbers of cells and the size of the trachea were actually reduced. Perturbation of the cell cycle therefore alters the normal balance of proliferation and growth characteristic of this tissue although how this occurs was not explained. Curiously, G2 phase arrested Tr2 cells were refractory to both checkpoint inhibition and ectopic activity of Cdk1 earlier in larval development, apparently by a novel mechanism. I look forward to hearing about their future work on this aspect of tracheal development.

I found the revised manuscript considerably improved, with the logical flow of the experiments much easier to follow. The figures were clear and the data convincing.

*Reviewer #3:*

Overall, the authors present a very nice story about ATR/chk1-dependent G2 lengthening as an important mechanism in building the fly trachea. I have a number of comments about data presentation that should be addressed to make the manuscript suitable for publication.

The graph in Figure 1G is non-intuitive. Please re-configure. It appears that two different things are being plotted on the graph, and "DRAQ5 Intensity bins" as an axis label does not convey any clear meaning.

Figure 1I- Can the authors more directly address how it was determined that the pre-WL3 levels of mRNA are or are not significantly above the level of background detection? Stg RNAi seems like a good control to do here.

In several places, I could not find clear information on the N values for experiments, or any information on how many times an experiment was replicated. For example, Figure 1J. How many times were the control and stg RNAi experiments performed? Every experiment should have information on N values and replicates.

For Figure 1J, how was the difference in stg protein expression normalized between control and stg RNAi animals? Please provide some quantitative data with replicates for this experiment. Similar comment for the cdc2 staining in Figure 4.

How do we know that the ATM RNAi is a valid reagent, given the negative result? Has this RNAi line been validated in other publications?

The authors do not comment on the dramatic difference in cell # increase between ATR and chk1 RNAi. Why would ATR knockdown be less efficient? Can the authors comment on whether the ATR RNAi is less effective, or whether redundant chk1 regulators may be involved? If neither possibility can be ruled out, at least note the difference, as it is striking in Figure 2.

Figure 5 – the Γ H2AX staining experiment lacks a positive control (such as staining after X-irradiation). Also, it's not clear to me what the single labeled nuclei labeled by this mammalian antibody are detecting. Why not use γ H2AV instead?

In Figure 5A, for the WL3 RNAi condition, the DAPI image has grid lines over it. Can the authors provide a better image?

---

## [Author Response]

[…] Overall, the reviewers were convinced that the ATR-Chk1 checkpoint controls the cell cycle during tracheal development, and felt that this was an interesting use of the checkpoint in development that warrants reporting in eLife. However the reviewers were not as convinced by the data on mitochondrial regulation of the cell cycle, or growth control by Stg, and felt that more data would be required to support these aspects of the paper. Thus we'd like to suggest two options you might follow in revising the paper.First, we think you could remove the data on mitochondria and PINK1, and also the data on Stg and growth, and submit a shorter, simpler paper including only the data relevant to the ATR-Chk1 mechanism, which the reviewers felt was relatively strong. This is your simplest option and could probably be achieved in a reasonable timeframe.Second, an alternative would be to provide a revision with more data from new experiments addressing mitochondrial fission, PINK1, and Stg-mediated growth. In this case the reviewers would have to be convinced by the new data, and its relevance to the ATR-Chk1 checkpoint. I expect this will be significantly more complex and time-consuming than option #1.

We would like to take option (1). The revised manuscript has focused on the role of the ATR-Chk1 pathway in developmental G2 arrest. We have omitted the sections on mitochondrial remodeling and the role of Stg in growth.

Reviewer #1:[…] The interesting aspect for a general audience is that ATR/Chk1 are used here as developmental regulators rather than in DNA damage response, and there are not very many examples of this general mechanism. The paper stops short of determining how ATR/Chk1 and mitochondrial fission are regulated developmentally. This would of course be nice to know, but isn't essential to make this study valuable.

We thank the reviewer for the insightful and generous appraisal of our work. We are currently investigating the mechanisms that regulate ATR-Chk1 and mitochondrial fission-fusion by way of a genetic screen. The characterization of the hits that have emerged from this screen is underway but too preliminary to report here.

At the end of their paper the authors show some data (Figure 5-6) suggesting G2 arrest promotes growth, and that stg/Cdc25 promotes ribosome biogenesis and interphase growth of tracheoblasts, but this is very preliminary, not directly relevant to the paper's central message. I suggest that this data be removed, and the work on ATR/Chk1 presented a short report.

We agree and have revised the figures and the text to this effect.

1) Although the authors' analysis of the tracheoblast cell cycle is careful, they have not measured DNA. As a 4N DNA content is the best marker for G2 arrest, this should be done. The FUCCI markers could be misleading if the arrest mechanism is unusual (subsection “Thoracic tracheoblasts arrested in G2 express Cdc25/Stg and other mediators of G2-M”, second paragraph).

The analysis of DNA content of larval tracheal cells has been reported previously (Guha et al., 2008 and Sato et al., 2008). These studies show that the DNA content of mitotically competent Tr2 tracheoblasts decreases in the L2-L3 interval in contrast to the DNA content of endoreplicating tracheal cells that increases in the same interval. The analyses also show that the DNA content of cells in G2 arrest is higher than that of cells that are actively cycling but do not specifically address whether the DNA content of arrested cells is ~4N.

To address whether G2-arrested cells are ~4N we have compared the DNA content of proliferating Tr2 tracheoblasts in G1 and G2 and the DNA content of Tr2 tracheoblasts in L2 and 0-8 L3 h L3. The cells in G1/G2 at the wandering L3 stage were marked by FUCCI. A histogram showing the DNA content at all of the abovementioned stages is shown in Author response image 1. This histogram reveals that the Tr2 tracheoblasts in L2 or 0-8 L3 have a DNA content that is comparable to the DNA content of proliferating tracheoblasts in G2. We infer that the DNA content of the arrested tracheoblasts is ~4N. This data has been included in Figure 1 (Figure 1G) to substantiate our claim that the arrested cells are in G2.

**Author response image 1. respfig1:** Comparison of the DNA content of actively dividing cells in Tr2 DT in G1 and G2 at the wandering L3 stage (FUCCI) and cells in Tr2 DT at L2 and 0-8 h L3. Histogram shows percent nuclei from WL3 in G1 (green, n=13) or G2 (yellow, n=15), L2 (black, n=23) and 0-8 h L3 (grey, n=14) in each DRAQ5 intensity bin (DNA content). We have also repeated our BrdU incorporation studies in animals in L2, L3 and in animals fed BrdU through the L2-L3 molt. We do not detect any BrdU incorporation in Tr2 DT in any of these conditions.

2) Figure 1. Showing a schematic summary of the FUCCI data here, without showing real data, is not acceptable. I suggest that some part of the actual FUCCI data be included in the main figure.

We have revised Figure 1 to include FUCCI.

3) The authors argue that stg is expressed during the G2 arrest, and imply that the levels should be sufficient to trigger M phase. However the data don't really support this. Stg levels clearly rise as the cell begin to divide, and the lower levels may be sub-threshold. There is not data in the paper to rule this out, so I suggest the authors moderate their statements that Stg is present during G2 arrest. These statements, in the title, Abstract, and elsewhere are misleading.

We think our data is consistent with the possibility that levels of Stg mRNA/protein in cells that are arrested is sufficient for mitotic entry. As we have shown, levels of Stg mRNA/protein in cells at the time cells enter division (32-40 h L3) is not significantly higher than when the cells are arrested (L2, 0-8 h L3, 1624 h L3). The levels of Stg mRNA/protein increase after the cells enter mitosis. We have now also assayed the levels of Stg mRNA/protein in animals that express Chk1^RNAi^ and divide precociously at 16-24 h L3. We find that Stg mRNA/protein levels in these mutants at 16-24 h L3 are comparable to control (G2-arrested) at the same stage. Thus, we find no evidence for the upregulation of Stg at the time the cells enter mitosis. It is plausible that the levels of Stg in arrested cells is sufficient for mitotic entry and that the observed upregulation of Stg contributes toward rapid proliferation thereafter.

We think that Stg is expressed in cells in G2 arrest and that comments about the dynamics of Stg mRNA are not central to the narrative. Thus they have been omitted from the revised version.

With regard to revising the title and Abstract, we have now focused on the role of ATR-Chk1 in the context of G2 arrest and the possible developmental significance of the G2 arrest as inferred from the loss Chk1 (please see response to comment #8).

4) Figure 2. Tp bolster their analysis and cement their conclusions, the authors should quantify numbers of PH3 figures at timpanist in the WT and Chk1-RNAi and/or ATR-RNAi animals. The one Ph3 figure shown is not even obviously in a dividing cell, and the reliance on cell number counts raises some issues as cell death also occurs in this organ at this time.

We have quantified the frequency of pH3^+^ figures in wild type and Btl-Chk1^RNAi^ animals at 0-8 h L3. While Tr2 DT in wild type animals have NO pH3^+^ figures at this stage (0%, n=24 tracheae), 20% of Tr2 DT in Btl-Chk1^RNAi^ animals have pH3^+^ figures (n=30 tracheae). We infer that there is precocious mitotic entry in Btl-Chk1^RNAi^ animals at 0-8 h L3 and this leads to increased numbers of cells at 16-24 h L3. This data has been included in the text.

5) Figure 3 shows some very raw data that is not obviously conclusive. I suggest the authors design a better experiment to test their idea and present that instead. As is, this figure doesn't add much to the paper. Clonal analysis might be useful here.

The reason this data has been included in the paper is it speaks to the roles of Chk1 in the regulation of mitotic entry and in the regulation of the rate of proliferation post mitotic entry. We have clarified this in the text and reformatted the figure to communicate what is strictly necessary.

6) Better data are required to document the onset of mitochondrial fission in L3. Ideally this data will be quantitative. They should also provide data showing that PINK1 promotes mitochondrial fission in their system.

We appreciate these are important concerns.

Our intention is to communicate that G2 arrest in tracheoblasts involves the coordination of genetically distinct processes and that Chk1 activation is but one of these. Thus, we have included data concerning responsiveness (or lack thereof) to Cdc2^AF^ overexpression. Since the data on the timecourse and regulation of mitochondrial architecture is not strictly necessary it has been omitted.

7) The additional mitoses generated by promoting mitochondrial fission by PINK1 are a relatively few. The authors make a lot of this small effect, but it's not clear that the combination of Chk1 loss and PINK1 completely relieves G2 arrest. From Figure 4F it looks like it does not. Perhaps the induction of Stg in L3 is important, or some other mechanism. The authors should take this into account in their Discussion.

We recognize that this is an important concern. Please see response to comment#6.

8) The section on growth in G2 is rather descriptive, and the effects of PINK1 and Chk1-RNAi are small and perhaps insignificant. Looking at this data, I would conclude that PINK1 may inhibit growth, not that G2 arrest promotes it. I suggest removing this inconclusive data (Figure 5).

We think that G2 arrest may be necessary for requisite tracheal growth. The three pieces of evidence that lead us to think in this manner are as follows:

a) Maximal cell and organ growth occurs while the cells are in G2 arrest (Figure 6AC).

b) The loss of Chk1 leads to precocious mitotic entry and perturbs growth such that Tr2 DT does not grow to the requisite size. The data on the role of Chk1 in growth has been included in the revised manuscript (Figure 6D).

c) That the overexpression of Chk1 prolongs G2 arrest and leads to increased tracheal size (Figure 6D).

We recognize that these data do not reveal whether arrest in G2 directly or indirectly (e.g. via regulation of cell fate) contributes toward cellular and organ growth. Regardless, we think that the association between arrest in G2 and growth is fairly clear and that our genetic data suggests a functional role for this type of cell cycle arrest.

9) As noted above, I suggest removing the data shown in Figure 6, since it is very preliminary and not directly relevant to the paper's central message.

We agree that the data pertaining to the role for Stg in growth is not relevant and has been omitted from the revised manuscript.

Reviewer #2:[…] The manuscript was clearly written and the experiments appear to have been carefully controlled and quantified. Many of the figures were difficult to view however (horizontal orientation).

We regret the inconvenience caused by horizontal orientation.

Reviewer #3:[…] Using assays for protein and mRNA expression along with genetics, they convincingly show that ATR/Chk1 and Stg are both present during the prolonged G2, and that Chk1 represses premature mitotic entry. Also convincing are the data supporting cell growth during prolonged G2.Less convincing is the authors' argument that mitochondrial dynamics are important for G2 arrest. The authors present images of mitoGFP as evidence of developmental changes in mitochondrial dynamics, which they argue are altered when Cdc2 activity is manipulated. I find these data unconvincing and can't tell what is being assayed, and there is no quantitation of such data to speak to reproducibility. EM may be needed to more directly assay mitochondrial dynamics. Further, overexpression of a single mitochondrial regulator (pink1) is presented as additional evidence of the role of mitochondria in cell cycle regulation. In addition to being an ectopic gene expression experiment, the effects are relatively minor, and it's unclear whether any effects on cell cycle/cell size are in any way directly altering cell cycle regulation. Messing with mitochondria may disrupt cell fate, which could cause premature cell cycle entry, for example. Additional genetic evidence of a specific role for mitochondrial dynamics is definitely needed.

In the revised manuscript we have focused on the role of the ATR-Chk1 axis in developmental G2 arrest and omitted the sections on mitochondrial architecture and the role of Stg in growth.

I was also confused by the author's interpretation of the loss of cells in Chk1 RNAi animals at wandering L3, which they attribute entirely to long mitosis, as opposed to premature mitotic entry. First of all, they directly contradict this interpretation on line 243, where they say that Chk1 loss leads to earlier than normal mitotic entry. Second, longer mitosis on its own should not lead to drastic changes in cell number, as mitosis is usually a very small fraction of the cell cycle. Perhaps in the absence of chk1 there are aberrant mitoses, which lead to cell elimination by a non-apoptotic means, as is widely reported in numerous systems? This possibility is not discussed.

In the first paragraph of the subsection “Fused, not fragmented, mitochondria contribute to G2 arrest in tracheoblasts”, we state the observation that cells expressing Chk1^RNAi^ enter mitosis precociously. If these cells were to proliferate at the same rate as in control animals we would expect supernumerary cells at wandering L3. However, this is not the case. We think that Chk1 has two distinct roles in the regulation of cell proliferation in Tr2. First, Chk1 prevents precocious mitotic entry. Second, Chk1 enables faster mitoses. There is a precedence for the latter in cultured cells and in other contexts and our data is consistent with this.

[Editors' note: further revisions were requested prior to acceptance, as described below.]

Reviewer #1:The revision of this paper follows the plan agreed upon after the first review, namely to simplify the paper by deleting some weak data and focus the presentation on the novel, interesting cell cycle effects that were discovered during tracheal development. The paper is now concise and largely convincing in its conclusions. However it still has six figures, and I suggest that some of these could be combined, since they are small. The quality of the data has also been improved with some additions as suggested in the reviews. I think the paper is essentially appropriate for publication in this form. However upon reading it, I did note several instances where the data, or the presentation thereof, could be improved. I invite the authors to address these specific points with a further round of minor revisions.

We would prefer to leave the main figures unchanged if that is permissible.

1) For Figure 1, it would be very helpful to provide a graph of PH3+ cell counts at the different stages shown with the other cell cycle markers.

A graph showing frequency of pH3^+^ cells/tracheal segment (mitotic index) at different larval stages has been included in Figure 1—figure supplement 1. This graph shows that the cells in Tr2 DT are quiescent in L1, L2, 0-8 h L3, 16-24 h L3 and mitotically active 3240 h L3 onwards. This is consistent with the timecourse of cell numbers shown in Figure 1I. The text has been revised accordingly.

2) Also for Figure 1, the graph showing DNA contents (Figure 1G) is rather confusing. I suggest graphing this data as in Figure 1I, with stage on the X axis and% cells with different ploidies on the Y axis.

The data is now presented in three separate panels. We hope that this representation clearly shows that the DNA content of arrested cells in L2 and early L3 is comparable to the DNA content of proliferating cells in G2 at WL3.

3) For Figure 2, we should see an increase in PH3+ cells in the Btl>ATR-RNAi and Btl>Chk1-RNAi samples. Please count the numbers of PH3+ cells for these genotypes and provide graphs. The data as is is substandard.

A graph showing the frequencies of pH3^+^ cells/tracheal segment in Btl-ATR^RNAi^ and BtlChk1^RNAi^ in L2, 0-8 h L3, 16-24 h L3 has been included in Figure 2 (Figure 2E). Please note that the timecourse of pH3^+^ cells in wild type has been added to Figure 1—figure supplement 1.

4) For Figure 6D, the authors should add p values for the growth increases and decreases. Pictures of the larger and smaller trachea would also be nice (but not essential.

A statement regarding statistical significance and a reference to the raw data/statistical analysis in Figure 6—source data 1 has been added.

5) Regarding Figure 6 and the statement at the beginning of the Discussion, I believe the Btl>Chk1-RNAi trachea have fewer cells due to effects earlier in development (noted earlier in the paper). This reduction in cell numbers might explain their smaller size. The authors should consider this problem with their interpretation and provide additional data using the Btl-TS driver if necessary. If no new data is provided the conclusion may have to be changed.

The numbers of cells in Tr2 DT in wild type and Btl-Chk1^RNAi^ animals are comparable starting out (L2, 0-8 h L3, see Figure 2D). At 32-40 h L3, the numbers of cells in Btl-Chk1^RNAi^ tracheae are higher than wild type as the cells in these tracheae enter mitoses earlier (see Figure 2). Shown in Figure 6 are the sizes of Tr2 DT in wild type and Btl-Chk1^RNAi^ animals at 32-40 h L3, a stage at which there are more cells in Btl-Chk1^RNAi^ than in wild type.

We have also included in Figure 6—figure supplement 1 a plot of the sizes of Tr2 DT in wild type and Btl-Chk1^RNAi^ animals at 0-8 h L3. Together, the sizes of tracheae at 0-8 h L3 and at 32-40 h L3 show that in Chk1 mutant animals the growth of Tr2 DT in the early L3 period is what is affected.

6) Please provide a citation for the statement in the last paragraph of the subsection “Fused, not fragmented, mitochondria contribute to G2 arrest in tracheoblasts”.

The citations Xiao et al., 2003 and O'Connell et al., 1997 have been added to the text.

Reviewer #3:Overall, the authors present a very nice story about ATR/chk1-dependent G2 lengthening as an important mechanism in building the fly trachea. I have a number of comments about data presentation that should be addressed to make the manuscript suitable for publication.The graph in Figure 1G is non-intuitive. Please re-configure. It appears that two different things are being plotted on the graph, and "DRAQ5 Intensity bins" as an axis label does not convey any clear meaning.

The data is now presented in three separate panels. We hope that the data now clearly shows that the DNA content of cells in L2 and early L3 is comparable to the DNA content of proliferating cells in G2.

Figure 1I- Can the authors more directly address how it was determined that the pre-WL3 levels of mRNA are or are not significantly above the level of background detection? Stg RNAi seems like a good control to do here.

The goal of the qPCR analysis was to determine if there is a transcriptional upregulation of Stg at the time of mitotic entry. Our analysis shows that there is no significant increase in the levels of Stg mRNA at the time of mitotic entry. We have not tried to determine whether the pre-WL3 levels of Stg mRNA are or are not significantly above “background”. All we would like to show here is that we do not detect a significant increase in levels of expression at the time the cells enter mitosis.

In several places, I could not find clear information on the N values for experiments, or any information on how many times an experiment was replicated. For example, Figure 1J. How many times were the control and stg RNAi experiments performed? Every experiment should have information on N values and replicates.

We have revised the text to include information about N values for all experiments and immunostaining experiments in particular. The staining shown in Figure 1J is representative of 3 independent immunohistochemical experiments in which at least 9 tracheae from each of the conditions shown were stained, examined by eye and then taken for imaging (stages are L2, 0-8 h L3, 32-40 h L3, Wandering L3 StgRNAi L2, and secondary alone control Wandering L3).

For Figure 1J, how was the difference in stg protein expression normalized between control and stg RNAi animals? Please provide some quantitative data with replicates for this experiment. Similar comment for the cdc2 staining in Figure 4.

The stained samples were mounted in ProLong Diamond with DAPI and imaged using a confocal microscope on the same day using the same imaging parameters. The laser power and gains were set in each experiment using specimens stained with the secondary antibody alone as the baseline. Any reported change in the level of staining is relative to level of staining in tracheae incubated with the secondary antibody alone. Author response image 2 shows the average intensity values for Stg staining at L2, 0-8 h L3 and WL3 from 5 Tr2 tracheae each and 3 tracheae from StgRNAi (intensity measurements were made on flattened z-stacks of Tr2 tracheae using Image J). The images used for the quantitation in Author response image 2 were randomly picked from our sample set.

**Author response image 2. respfig2:** 

The staining for Cdc2 and CycB shown in Figure 1—figure supplement 2 is representative of 2 independent immunohistochemical experiments in which at least 6 tracheae from each of the conditions shown were stained and imaged as for Stg (L2, Cdc2/CycB RNAi L2, and secondary alone control L2). In addition to this we have also examined levels of Cdc2 and CycB in 6 tracheae each at 0-8 h L3, 32-40 h L3 and Wandering L3 in 2 independent experiments though this data has been shown.

The staining for Cdc2 shown in Figure 4 is representative of 3 independent immunohistochemical experiments in which at least 5 tracheae from each of the conditions shown were analyzed (L2 hs/no-hs Cdc2AF, wild type wandering L3, no primary wandering L3).

How do we know that the ATM RNAi is a valid reagent, given the negative result? Has this RNAi line been validated in other publications?

The same line has been validated in another study (cited below) and this reference has been added to the text.

Kerry Flegel, Olga Grushko, Kelsey Bolin, Ellen Griggs and Laura Buttitta (2016). Roles for the Histone Modifying and Exchange Complex NuA4 in Cell Cycle Progression in *Drosophila melanogaster* GENETICS 203(3), 1265-1281.

The authors do not comment on the dramatic difference in cell # increase between ATR and chk1 RNAi. Why would ATR knockdown be less efficient? Can the authors comment on whether the ATR RNAi is less effective, or whether redundant chk1 regulators may be involved? If neither possibility can be ruled out, at least note the difference, as it is striking in Figure 2.

We have noted these possibilities in the text.

Figure 5 – the Γ H2AX staining experiment lacks a positive control (such as staining after X-irradiation). Also, it's not clear to me what the single labeled nuclei labeled by this mammalian antibody are detecting. Why not use γ H2AV instead?

While the γ H2AX antibody that we have used is raised against a conserved Serine residue at the C-terminal of the mammalian H2AX protein, antibodies raised against this pSer have been shown to cross react with *Drosophila* γ-H2AV (see Rogakou et al., 1999). Thus, the γ H2AX antibody is likely detecting γ-H2AV in our samples.

Antisera against γ-H2AX antibody have previously been used to detect DNA damage in *Drosophila*. Bayer and colleagues (Bayer et al., 2017) have shown larvae exposed to X-ray irradiation undergo DNA damage (see Bayer et al., 2017). The same study reported that levels γ-H2AX staining in larval tissues exposed to X-tray irradiation (40 Gy) were higher than in controls. As part of our initial characterization of the gammaH2AX antibody that we have used in this study, we subject wandering L3 larvae to X-ray irradiation (20Gy, 40 Gy) and stained imaginal discs from treated larvae 1 h post irrradiation with the antibody. We observed that the frequency of γ H2AX^+^ cells increased from 20 Gy to 40 Gy (see Author response image 3). We concluded that the γ-H2AX antibody was detecting DNA damage in *Drosophila* cells. We have added a description of the positive control to the text.

**Author response image 3. respfig3:** 

The occurrence of γ-H2AX^+^ nuclei in control animals at wandering L3 is intriguing. It has been reported that cells undergoing apoptosis also phosphorylate the Ser-139 on H2AX (see Rogakou et al., 2000). Thus, it is plausible that some of the γ-H2AX^+^ nuclei in control and Btl-Chk1^RNAi^ tracheae at Wandering L3 are in fact apoptotic cells. Consistent with this, we note that the frequencies of γ-H2AX^+^ nuclei and activated-Caspase3^+^ cells in control and Btl-Chk1^RNAi^ are comparable (Control: γ-H2AX^+^=1.57 ± 0.53 (n=7), activated Caspase3+=2.77 ± 1.17 (n=13); Btl-Chk1^RNAi^: γ-H2AX^+^=1.43 ± 0.53 (n=7), activated Caspase3+=2.3 ± 0.85 (n=13)). We have mentioned this in the text.

In Figure 5A, for the WL3 RNAi condition, the DAPI image has grid lines over it. Can the authors provide a better image?

This has been rectified.